# Reaction performance prediction with an extrapolative and interpretable graph model based on chemical knowledge

Shu-Wen Li ●[1], Li-Cheng Xu[1], Cheng Zhang[2], Shuo-Qing Zhang ●[1] ✉ & Xin Hong ●[1,3,4] ✉

Accurate prediction of reactivity and selectivity provides the desired guideline for synthetic development. Due to the high-dimensional relationship between molecular structure and synthetic function, it is challenging to achieve the predictive modelling of synthetic transformation with the required extrapolative ability and chemical interpretability. To meet the gap between the rich domain knowledge of chemistry and the advanced molecular graph model, herein we report a knowledge-based graph model that embeds the digitalized steric and electronic information. In addition, a molecular interaction module is developed to enable the learning of the synergistic influence of reaction components. In this study, we demonstrate that this knowledge-based graph model achieves excellent predictions of reaction yield and stereoselectivity, whose extrapolative ability is corroborated by additional scaffold-based data splittings and experimental verifications with new catalysts. Because of the embedding of local environment, the model allows the atomic level of interpretation of the steric and electronic influence on the overall synthetic performance, which serves as a useful guide for the molecular engineering towards the target synthetic function. This model offers an extrapolative and interpretable approach for reaction performance prediction, pointing out the importance of chemical knowledge-constrained reaction modelling for synthetic purpose.

The chemical comprehension and accurate prediction of reactivity and selectivity provide the foundation for the rational and efficient exploration of massive synthetic space[1,2]. This establishment of the structure–performance relationship (SPR) has been focused on the reaction mechanism study and elucidation of the determining transition state model[3]. Using the transition state model, chemists can elucidate the origins of the observed reactivity/selectivity trend and make synthetic judgments based on chemical theory and empirical experience[4]. This classic knowledge-driven strategy has reached remarkable success in synthetic chemistry and continues to provide strong support for the discovery of new catalysts, reagents, and reaction[5]. Despite the advantage of offering qualitative guidance in the synthetic universe, it is challenging for the knowledge-driven strategy to handle the high-dimensional SPR without a clear mechanistic basis and analytic equation. The seemingly subtle change in catalyst, additive, or even solvent may result in significant perturbation of the overall synthetic performance[6,7]. This is why laborious and repetitive condition

[1]Center of Chemistry for Frontier Technologies, Department of Chemistry, State Key Laboratory of Clean Energy Utilization, Zhejiang University, Hangzhou 310027, China. [2]Department of Chemistry, University of Science and Technology of China, Hefei, China. [3]Beijing National Laboratory for Molecular Sciences, Zhongguancun North First Street No. 2, Beijing 100190, PR China. [4]Key Laboratory of Precise Synthesis of Functional Molecules of Zhejiang Province, School of Science, Westlake University, 18 Shilongshan Road, Hangzhou 310024 Zhejiang Province, China. ✉e-mail: angellasty@zju.edu.cn; hxchem@zju.edu.cn

optimization is still inevitably required, limiting the efficiency of synthetic development[8].

The data-driven approach has recently emerged as a powerful strategy for SPR establishment[9,10]. By harnessing the interrelationship within the synthetic data, modern machine learning (ML) algorithms can create powerful models for synthetic prediction. Accurate predictions of reaction yield[11–14], kinetic rate[15,16] and activation energy[17–19], chemo-[20], regio-[21–25], and stereoselectivity[26–32]. have been achieved in a wide array of organic transformations, which validated the exciting concept of ML prediction of synthetic performances. However, the ML prediction and design of synthetic transformation are still far from mature. One of the major bottlenecks is the availability of the molecular encoding approach and the ML framework that are suitable for SPR prediction (Fig. 1a). Quantum chemical descriptors[27] are known for their solid physical basis and high descriptive ability, but their application typically requires a sophisticated understanding of the underlying reaction mechanism, and the descriptor generation can be time- and resource-consuming for large-scale screening. The string- and topological structure-based encodings (i.e., SMILES, molecular fingerprints, etc.)[33,34] do not require expert knowledge of the studied transformation and can be efficiently generated, while it is difficult to trace the physical organic origins of the synthetic performance. In addition, the extrapolation problem presents additional challenges for SPR prediction[35,36]. Current synthetic models still lack sufficient guidance for developing new catalysts and transformations.

Beyond the human-specified engineering of molecular encodings, the field of chemical prediction has been witnessing a growing interest in representation learning. Through the innovation and application of representation learning, data-driven predictions of molecular property[37,38] and reaction performance[39,40] have achieved significant progress. Particularly in SPR prediction, Jensen, Green, Coley, and co-workers combined the classic graph neural network (GNN) model with selected quantum chemical descriptors of reaction sites, developing a fusion model called QM-GNN[22,41] (Fig. 1b). This fusion model embeds site-specific electronic information into ML modeling, which improves the predictive ability towards regioselectivity[22] and reactivity[41] across a series of transformations. The success of the QM-GNN model indicated that enhancing the expression of local chemical information can provide valuable support for constraining synthetic modeling. Inspired by this model, we surmise that the SPR prediction can be further improved by enriching the local encodings of the chemical environment and strengthening the information interaction between reaction components.

In this work, we report a reaction performance model with two innovative designs (Fig. 1c): the knowledge-enhanced molecular graph provides an unbiased way to embed the digitalized steric and electronic information of the atomic environment, which enriches the representation of the entire molecule instead of specifying the controlling sites; the molecular interaction module allows the effective learning of the synergistic control by multiple reaction components, which enables the effective extension of molecular modeling to the realm of SPR modeling. This model achieves excellent yield and stereoselectivity predictions in a series of challenging tasks, and our additional experimental tests of asymmetric thiol addition of imines verify its extrapolation ability in new catalyst predictions. This accurate, extrapolative, and interpretable model provides a useful approach for reaction performance prediction, which accelerates the ML design of molecular synthesis.

## Results

### Encoding chemical information in the molecular graph

The key design of our knowledge-enhanced molecular graph is to embed the atomic information of the steric and electronic environment in the node. This introduces external chemical knowledge to improve the model's differentiation ability from the local chemical environment. The generation workflow of the designed molecular graph model, called steric- and electronics-embedded molecular graph (SEMG), is presented in Fig. 2 using 1-chloro-4-(trifluoromethyl)benzene as an example. The first step is to generate a molecular graph with a series of empty vertices from SMILES (Fig. 2a). Subsequently, the molecule is optimized under the GFN2-xTB[42] level of theory, and the digitalization of the local steric environment is realized using a spherical projection method developed by our group in previous work[43] (Fig. 2b). This approach, called spherical projection of molecular stereo-structure (SPMS), maps the steric environment by projecting the distance between the molecular vdW surface and a customized sphere from a designated center (exampled by chlorine in this case). Equirectangular projection of the mapped distance sphere creates a two-dimensional distance matrix, which is used as the embedded steric information for the graph vertex. For the embedding of the local electronic environment, the B3LYP/def2-SVP-computed electron density is used (Fig. 2c). This provides a reasonable estimation of the electron density distribution in real space, which supports the evaluation of the electronic environment neighboring the selected atom. Centered at the selected atom, a cube with the vdW diameter as the side length was divided into $7 \times 7 \times 7$ grids. The computed electron density values were recorded as a $7 \times 7 \times 7$ tensor, which is used as the embedded electronic information for the vertex of the graph model. Repeating the steric and electronic embeddings for each atom leads to the final SEMG for model training. To ensure the physical accuracy of the optimized geometry and the computed electron density, we performed careful benchmarking of a series of theoretical approaches, GFN2-xTB optimization and B3LYP/def2-SVP calculation of electron density was found to provide reasonable model inputs with solid physical accuracy and affordable computational cost for large-scale screening. Additional details of the information embedding are provided in the Supplementary Information.

### Design of molecular interaction GNN

Building upon the rich chemical information of SEMG, next, we modified the framework of GNN to make it suitable for SPR prediction. Particularly, a molecular interaction module is developed to enhance the information exchange between reaction components during the model training. The synergistic molecular interaction is of importance for the determination of SPR, while was rarely explored for the model designs of synthetic predictions; an interesting design from Liu and co-workers applied hyper-graph to enhance the intermolecular information exchange[44] which aimed to address the same issue as in our interaction module. Different from hyper-graph, our design of the molecular interaction GNN (MIGNN) relies on matrix operation to enable the information exchange (Fig. 3). The SEMGs of the reaction components are processed by the attention layer, in which the weight values are trained to capture the atomic contribution for reaction performance determination. Subsequent linear, convolution, max pooling, and flatten layers lead to a one-dimensional reaction vector. This reaction vector, which is the processed reaction representation with uneven local attention, is duplicated into two copies. One copy of the reaction vector undergoes the interaction module to enhance the information exchange between reaction components. In the interaction module, the matrix multiplication of the reaction vector allows the information of each reaction component to interact with those of the other components, whose details are further elaborated in the Methods section. This creates an interaction matrix, which is further processed by sequential attention, convolution, and flattening layer to produce an interaction vector with the

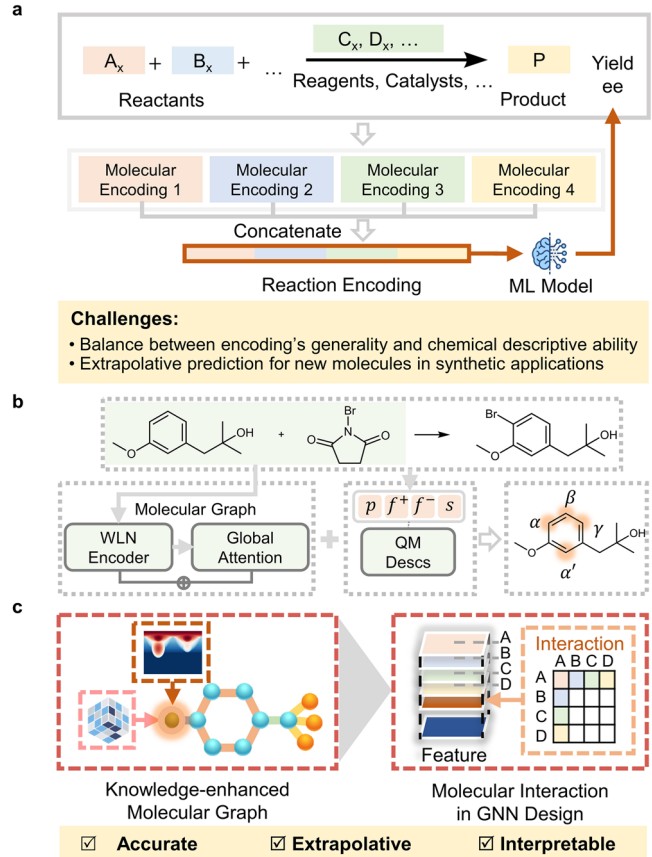

**Fig. 1 | Machine learning prediction of synthetic performance and molecular property. a** Representative strategy of synthetic prediction by concatenating the molecular encodings of reactant 1 (orange), reactant 2 (blue), additive (green), and product (yellow). **b** Previous work of quantum chemistry-augmented graph model for synthetic prediction using the WLN (Weisfeiler-Lehman network) encoder and QM (Quantum Chemistry) descriptors. The green background represents the input compounds and the architecture of the model. The orange background represents four quantum chemical descriptors. $\alpha$, $\beta$, $\gamma$, and $\alpha'$ indicate the reaction sites. **c** Chemical knowledge-based design of graph model for synthetic performance prediction (this work).

ability to focus on certain interacting pairs. The interaction vector and the other copy of the reaction vector are then concatenated as the final one-dimensional vector that represents the synthetic transformation, which passes the last attention and linear layers to provide the prediction value.

The interaction module of MIGNN provides an opportunity for the ML model to capture the synergistic effect of reaction components, which is challenging for the conventional ML framework of SPR prediction. Current ML models of synthetic transformation generally concatenate the molecular encodings in an unphysical sequence. This simple concatenation directly mixes the molecular encodings of all reaction components; thus, the physical information of intra- and intermolecular boundaries no longer exists, making it difficult for the ML model to directly capture the synergistic influence of multiple reaction components. In MIGNN, the interaction matrix allows all possible combinations of the physically meaningful reaction components to fully exchange their encoded information, which supports the model's prediction in the high-dimensional synthetic space with intertwined reaction components.

## Performance of the SEMG-MIGNN model

We next evaluated the predictive ability of the SEMG-MIGNN model. Both yield and enantioselectivity predictions were tested using Doyle's

dataset[11] of Pd-catalyzed C–N cross-coupling reactions between 4-methylaniline and aryl halides as well as Denmark's dataset[31] of chiral phosphoric acid (CPA)-catalyzed thiol addition to N-acylimines. These high-quality datasets provide valuable statistics that map the complete synthetic space given the studied reaction components, which have been widely applied as benchmark datasets in ML studies of SPR[13,40,45]. The data size and variations of the involved reaction components are described in Fig. 4a and Fig. 5a, respectively. To highlight the effectiveness of the chemical information embedding and the interaction module in the SEMG-MIGNN model design, we compared SEMG with the classic molecular graph[46] (baseline MG) which uses the limited atomic descriptions in the vertex (atom type, atomic number, binary definition of acceptor/donor, etc.). The MIGNN framework was compared to the classic GCN[47] designs without the interaction module. These variations together led to four possible modeling approaches: baseline MG-GCN, SEMG-GCN, baseline MG-MIGNN, and SEMG-MIGNN. Further technical details of the tested models are provided in the Supplementary Information.

Doyle's C–N coupling yield dataset was randomly split into 70% (training) and 30% (test), and ten trials of yield prediction were performed. For each model, a representative regression performance is elaborated in Fig. 4b. The baseline MG-GCN model gave unsatisfying prediction results; the $R^2$ value is 0.545, and the RMSE is 18.40%. Changing the baseline MG to the chemical information-embedded SEMG improves the regression performance, the SEMG-GCN model achieved a $R^2$ of 0.592 and RMSE of 17.56%. To our satisfaction, the training with MIGNN significantly improves the predictive ability of graph representation. Even with the baseline MG, the baseline MG-MIGNN model can achieve an excellent yield prediction with an average $R^2$ of 0.921 and RMSE of 7.69%. This highlights the synergistic yield control of the reaction components in the Pd-catalyzed C–N cross-coupling and the ability of the MIGNN framework to capture this effect. The usage of the SEMG-MIGNN model further improves the prediction performance, which is the best among the tested four combinations. The $R^2$ and RMSE of the representative SEMG-MIGNN modeling are 0.969 and 4.81%, respectively. In addition to the changes in graph representation and GNN framework, we also compared the SEMG-MIGNN model with other state-of-the-art (SOTA) ML approaches (Yield-BERT[40], DRFP[45], and MFF[13]) without the embedding of steric and electronic information. Table 1 presents the inter/extrapolative prediction performances of these models with varied splittings of the dataset. In the interpolation tasks with different ratios of training data, all the SOTA models were able to provide satisfying prediction performances, with SEMG-MIGNN showing limited improvements. In the extrapolation tasks, however, SEMG-MIGNN demonstrated noticeable advantages. We performed scaffold splitting[22,48] based on the structural variations of the compounds involved in the yield dataset, resulting in four extrapolation challenges of aryl halide, additive, ligand, and base; the details of these splittings are provided in the Supplementary Information (Supplementary Fig. 20). The tested SOTA models met difficulties in these extrapolation tasks with RMSEs ranging from 18% to 26%, making predictions with limited synthetic value. SEMG-MIGNN model can provide accurate extrapolative predictions for additives and ligands, with RMSE of 10.36% and 11.02%, respectively. The aryl halide and base extrapolations are more challenging, yet the SEMG-MIGNN model still achieved markedly superior performance compared to the other models. Detailed regression performances are provided in the Supplementary Information (Supplementary Fig. 21). These results further highlighted the SEMG-MIGNN model's ability to make extrapolative predictions for new molecules, emphasizing the significance of embedding chemical information in SPR modeling.

The evaluations of enantioselectivity prediction also verified the excellent ability of the SEMG-MIGNN model (Fig. 5). Denmark's

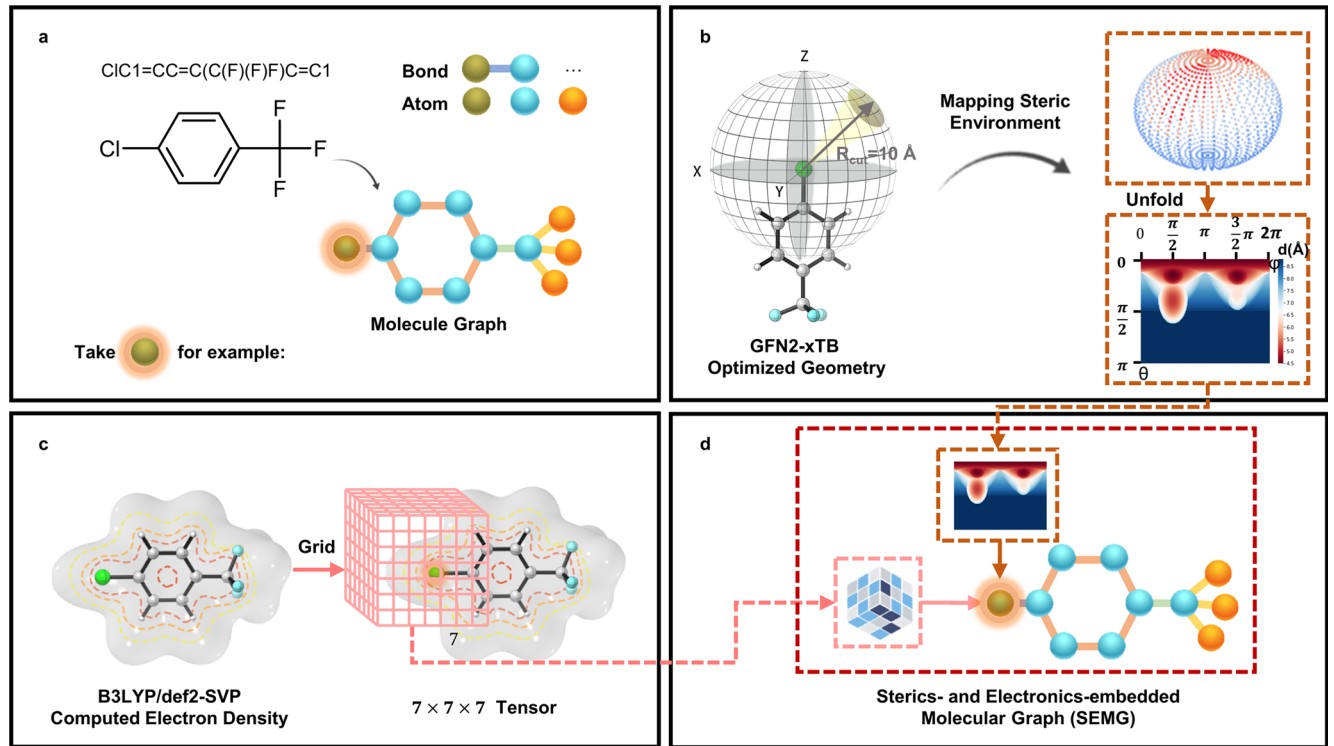

**Fig. 2 | Generation workflow of the steric- and electronics-embedded molecular graph (SEMG). a** Molecular graph generation from SMILES (simplified molecular input line entry system). **b** Embedding of the atomic information of the steric environment. **c** Embedding of the atomic information of the electronic environment. **d** Generated SEMG (steric- and electronics-embedded molecular graph) with embedded chemical information.

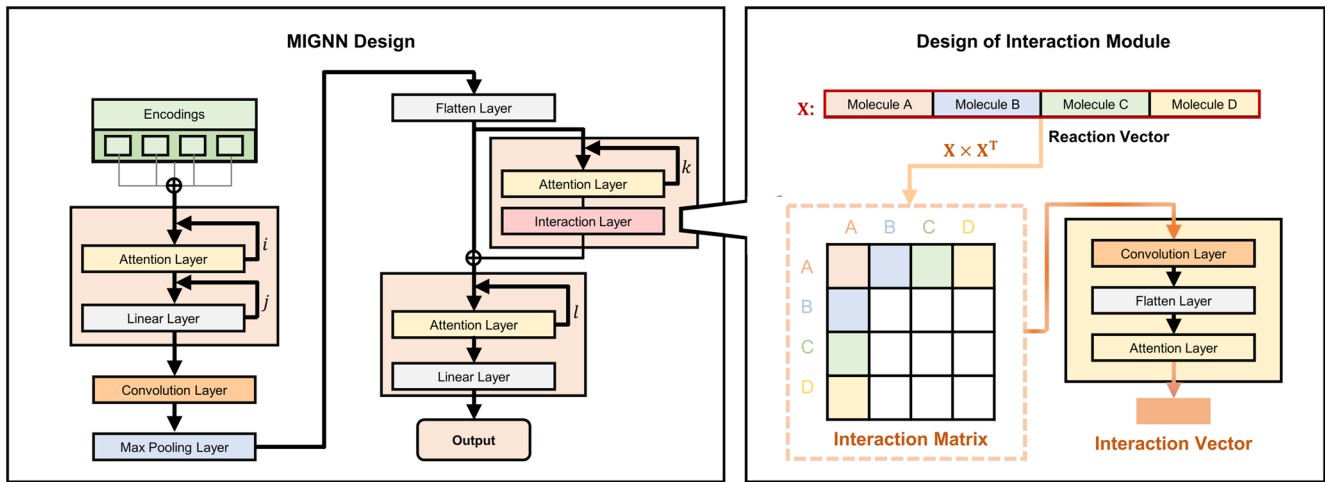

**Fig. 3 | Framework of molecular interaction graph neural network (MIGNN) and the design of the interaction module.** The MIGNN (molecular interaction graph neural network) first processes the SEMGs (steric- and electronics-embedded molecular graph) of the reaction components via the attention, linear, convolution, max pooling, and flatten layers, which leads to a one-dimensional reaction vector. Subsequently, the reaction vector is duplicated into two copies. One copy undergoes an interaction module that allows the information of each reaction component to interact with those of the other components, resulting in an interaction vector. This interaction vector and the other copy of the reaction vector are concatenated as the final one-dimensional vector, which passes the attention and linear layers to provide the prediction value.

asymmetric thiol addition dataset of 1075 transformations were randomly split into 600 (training) and 475 (test) following the data splitting in the original study[31] (Fig. 5a). Fig. 5b compares the GNN performances with or without the SEMG and MIGNN designs. Training with the baseline MG-GCN model again led to less satisfying regression performance; the representative trial has an $R^2$ of 0.778 and an RMSE of 0.332 kcal mol$^{-1}$. Changing either the graph representation or the GNN framework improved the predictive ability. The SEMG-GCN and baseline MG-MIGNN models have predictions of the enantioselectivity with

$R^2$ of 0.819 and 0.880, respectively. Consistently, the SEMG-MIGNN model is the best among the four tested combinations with an excellent prediction performance. The $R^2$ of the representative SEMG-MIGNN trial is 0.915, and the corresponding RMSE is 0.197 kcal mol$^{-1}$. The comparisons with other SOTA models further highlighted the predictive ability of the SEMG-MIGNN model in enantioselectivity tasks. Table 2 provides the inter- and extrapolative enantioselectivity predictions using SEMG-MIGNN and other SOTA models. In the interpolation tasks, all the SOTA models were able to provide accurate

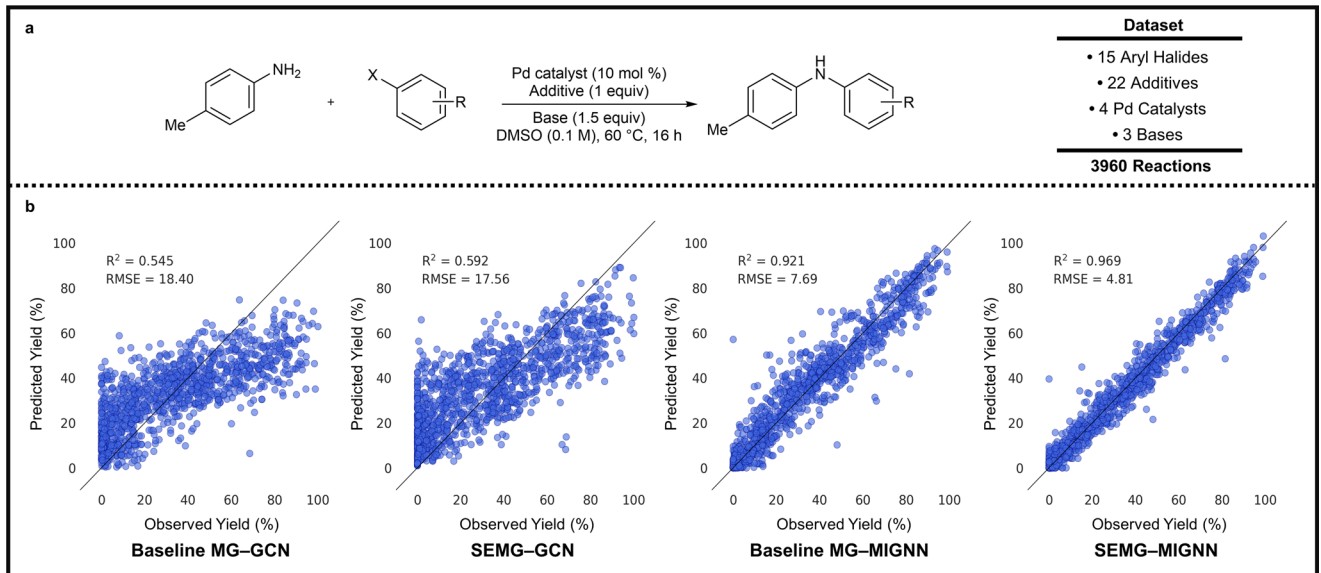

**Fig. 4 | Prediction of reaction yield by SEMG-MIGNN model (sterics- and electronics-embedded molecular graph with molecular interaction graph neural network). a** Overview of the dataset of Pd-catalyzed C−N cross-coupling reactions between 4-methylaniline and aryl halides. These data are originally published by Doyle and co-workers[11]. **b** Regression performances of baseline MG-GCN (baseline molecular graph with graph convolutional network), SEMG-GCN (steric- and electronics-embedded molecular graph with graph convolutional network), baseline MG-MIGNN (baseline molecular graph and molecular interaction graph neural network) and SEMG-MIGNN models (steric- and electronics-embedded molecular graph with molecular interaction graph neural network). The dataset was randomly split into 70% (training) and 30% (test). SEMG-MIGNN model (steric- and electronics-embedded molecular graph with molecular interaction graph neural network) outperforms the other tested combinations with $R^2$ and RMSE (root mean square error) of 0.969 and 4.81%, respectively. Source data are provided as data1.csv and Data_for_Fig_4.csv.

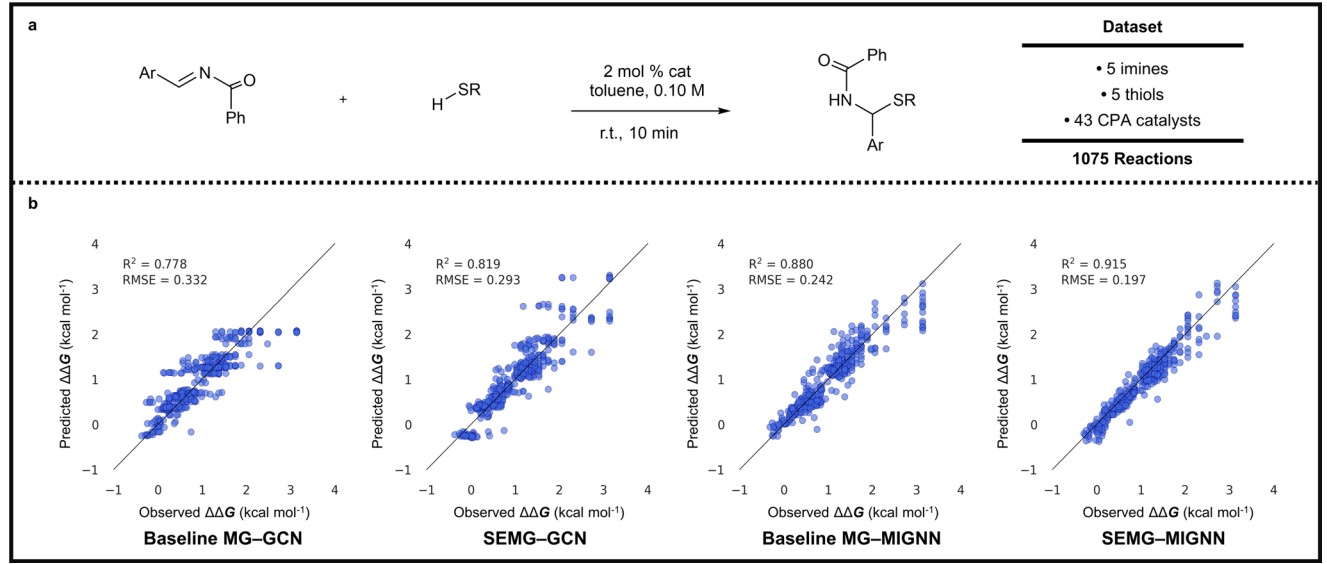

**Fig. 5 | Prediction of enantioselectivity by SEMG-MIGNN model (steric- and electronics-embedded molecular graph with molecular interaction graph neural network). a** Overview of the dataset of the chiral phosphoric acid-catalyzed thiol addition to *N*-acylimines. These data are originally published by Denmark and co-workers[26]. **b** Regression performances of baseline MG-GCN (baseline molecular graph and graph convolutional network), SEMG-GCN (steric- and electronics-embedded molecular graph with graph convolutional network), baseline MG-MIGNN (baseline molecular graph with molecular interaction graph neural network) and SEMG-MIGNN models (steric- and electronics-embedded molecular graph with molecular interaction graph neural network). The dataset was randomly split into 600 (training) and 475 (test) reactions. SEMG-MIGNN model (steric- and electronics-embedded molecular graph with molecular interaction graph neural network) outperforms the other tested combinations with $R^2$ and RMSE (root mean square error) of 0.915 and 0.197 kcal mol$^{-1}$ respectively. Source data are provided as data2.csv and Data_for_Fig_5.csv.

prediction performances; the SEMG-MIGNN model showed marginal improvement in all cases. In contrast, the thiol and catalyst extrapolation tasks differentiated the model performances. The SEMG-MIGNN model was still able to provide competent predictions with RMSE of 0.300 kcal mol$^{-1}$ (thiol) and 0.294 kcal mol$^{-1}$ (catalyst), which

outcompetes the tested SOTA models with marked differences. Detailed regression performances are provided in the Supplementary Information (Supplementary Fig. 23). Consistent with the yield predictions, the modeling of enantioselectivity prediction confirmed the excellent extrapolative ability of the SEMG-MIGNN model, providing

**Table 1 | Comparison of yield predictions between the SEMG-MIGNN model with other SOTA models**

| Data splitting | Yield-BERT | DRFP | MFF | SEMG-MIGNN |
|---|---|---|---|---|
| Random 90/10 | 5.20 ± 0.500 | 5.09 ± 0.500 | 6.34 ± 0.500 | **4.79 ± 0.500** |
| Random 70/30 | 5.82 ± 0.400 | 6.28 ± 0.300 | 6.77 ± 0.300 | **4.81 ± 0.400** |
| Random 50/50 | 7.62 ± 0.500 | 7.36 ± 0.300 | 8.55 ± 0.300 | **6.83 ± 0.500** |
| Random 30/70 | 9.41 ± 0.500 | 8.67 ± 0.500 | 10.09 ± 0.500 | **8.79 ± 0.700** |
| Aryl Halide[a] | 26.04 ± 0.300 | 26.19 ± 0.200 | 22.04 ± 0.200 | **19.34 ± 0.400** |
| Additive[a] | 21.29 ± 0.200 | 22.43 ± 0.200 | 21.66 ± 0.200 | **10.36 ± 0.200** |
| Ligand[a] | 20.04 ± 0.200 | 18.35 ± 0.200 | 18.85 ± 0.200 | **11.02 ± 0.200** |
| Base[a] | 19.40 ± 0.200 | 19.90 ± 0.200 | 20.66 ± 0.200 | **14.52 ± 0.200** |

Note: The best performance of each task is shown in bold. [a]These data splitting tasks refer to the extrapolative predictions based on the scaffold splitting of the reaction components. Details are elaborated in Supplementary Fig. 20. RMSEs are in %.

**Table 2 | Comparison of enantioselectivity predictions between the SEMG-MIGNN model with other SOTA models**

| Data Splitting | DRFP | MFF | SEMG-MIGNN |
|---|---|---|---|
| Random 90/10 | 0.190 ± 0.010 | 0.183 ± 0.010 | **0.180 ± 0.010** |
| Random 70/30 | 0.201 ± 0.010 | 0.212 ± 0.020 | **0.189 ± 0.010** |
| Random 50/50 | 0.248 ± 0.030 | 0.227 ± 0.030 | **0.205 ± 0.020** |
| Random 30/70 | 0.259 ± 0.030 | 0.243 ± 0.030 | **0.240 ± 0.020** |
| Imine[a] | 0.227 ± 0.005 | **0.226 ± 0.005** | 0.238 ± 0.005 |
| Thiol[a] | 0.774 ± 0.020 | 0.726 ± 0.020 | **0.300 ± 0.010** |
| Catalyst[a] | 0.565 ± 0.020 | 0.464 ± 0.020 | **0.294 ± 0.010** |
| Transformation[a] | 0.235 ± 0.005 | 0.264 ± 0.005 | **0.205 ± 0.005** |

Note: The best performance of each task is shown in bold. [a]These data splitting tasks refer to the extrapolative predictions based on the scaffold splitting of the reaction components. Details are elaborated in Supplementary Fig. 22. RMSEs are in kcal mol⁻¹.

strong support that the embedding of local chemical information is beneficial for SPR predictions.

## Experimental verifications of extrapolation ability

To further evaluate the predictive ability of the SEMG-MIGNN model and test its reliability in the application scenario of catalyst discovery, we next performed a series of experimental verifications using additional CPA catalysts in the asymmetric thiol addition to *N*-acyl amines. These transformations are the extensions to Denmark's enantioselectivity dataset, but the tested eleven CPAs are not present in the original dataset (Fig. 6). We also evaluated the structural differences between the tested CPAs and those in Denmark's dataset; the averaged correlation coefficient of Tanimoto similarity using Morgan molecular fingerprints is only 0.56 (Supplementary Fig. 18). The structural variation of these 11 CPA catalysts provides a reasonable experimental scenario to evaluate the model's extrapolative ability, which is essential for the ML-driven discovery of new catalysts and reagents. Training with Denmark's dataset, the SEMG-MIGNN model achieved excellent predictions for all the eleven tested cases (Fig. 6). The largest errors of prediction are only 0.21 kcal mol⁻¹ (**CPA-2, CPA-3**, and **CPA-11**), whose predictions are still synthetically useful. These additional experimental verifications corroborated the extrapolative ability of the SEMG-MIGNN model; the trained ML model is able to differentiate the performance of the candidate catalysts from related SPR data, making useful synthetic judgments without the requirement of a mechanistic model.

The experimental verifications also provided the opportunity to compare the predictive ability between the SEMG-MIGNN model and other ML approaches. Considering that DRFP and MFF models only contain topology-based encodings, we also evaluated the predictions of atom-centered symmetry functions (ACSFs) descriptors[49], which is a typical 3D descriptor for chemical modeling. The errors of predictions of these four models are compared in Fig. 7, in which the advantage of SEMG-MIGNN predictions is evident. It is worth noticing that all the SEMG-MIGNN predictions are within a reasonable error, while the other models tend to encounter pitfall cases where the error of prediction can be larger than 0.5 kcal mol⁻¹ and misleading for synthetic designs. Full details of the prediction results are provided in the Supplementary Information (Supplementary Table 18).

## Model interpretation

In addition to the improved predictive ability, another key merit of graph representation is the opportunity to trace the atomic contribution to the overall determination of the SPR prediction. This allows a chemically meaningful perspective to interpret the ML model and provide atomic-level insight for the structural engineering of catalysts and reagents. Because the embeddings of the steric and electronic information are independent in our SEMG design, the influence of the steric and electronic encodings on the overall synthetic performance prediction can be explored based on the perturbation of prediction upon eliminating the encodings. By setting the studied steric or electronic encodings to zero, the mathematical outcome directly related to these encodings would be deleted, which led to a perturbed prediction result. This perturbation provided a useful perspective of the steric and electronic effects on SPR. We want to emphasize that setting encodings to 0 is only a relatively intuitive operation, and the resulting perturbation is not additive. The reason for adopting this approach is that there is currently lack rigorous methods to analyze the causal relationship of highly parametrized nonlinear models. Fig. 8a shows the changes in predicted yield by eliminating the steric or electronic encodings using 3-chloropyridine as an example. In Doyle's dataset, 3-chloropyridine has 264 associated transformations. By removing the steric or electronic encodings of each atom from 3-chloropyridine, 4752 perturbations of the yield predictions were identified. In the yield prediction task, removing the steric encodings led to limited influence on the predicted values; 4031 transformations have a value change within 20% (Fig. 8a). In contrast, removing the electronic encodings resulted in a noticeable change in the reaction yield prediction. Over 3000 cases have a perturbation higher than 20% (Fig. 8a). This model interpretation confirmed the chemical knowledge that the electronic effect plays a dominant role in the reaction yield of Pd-catalyzed Buchwald−Hartwig reaction, while the steric effect is limited for the explored reactants. This chemical insight indeed followed the empirical mechanistic understandings, which indicated that the steric and electronic effects of the SEMG model can be analyzed to provide chemically meaningful understandings.

Not only the SEMG representation can quantitatively elaborate the steric and electronic effects on the prediction of synthetic performance, but the introduction of the attention layer also offers an atomic level precision to identify the key positions of performance

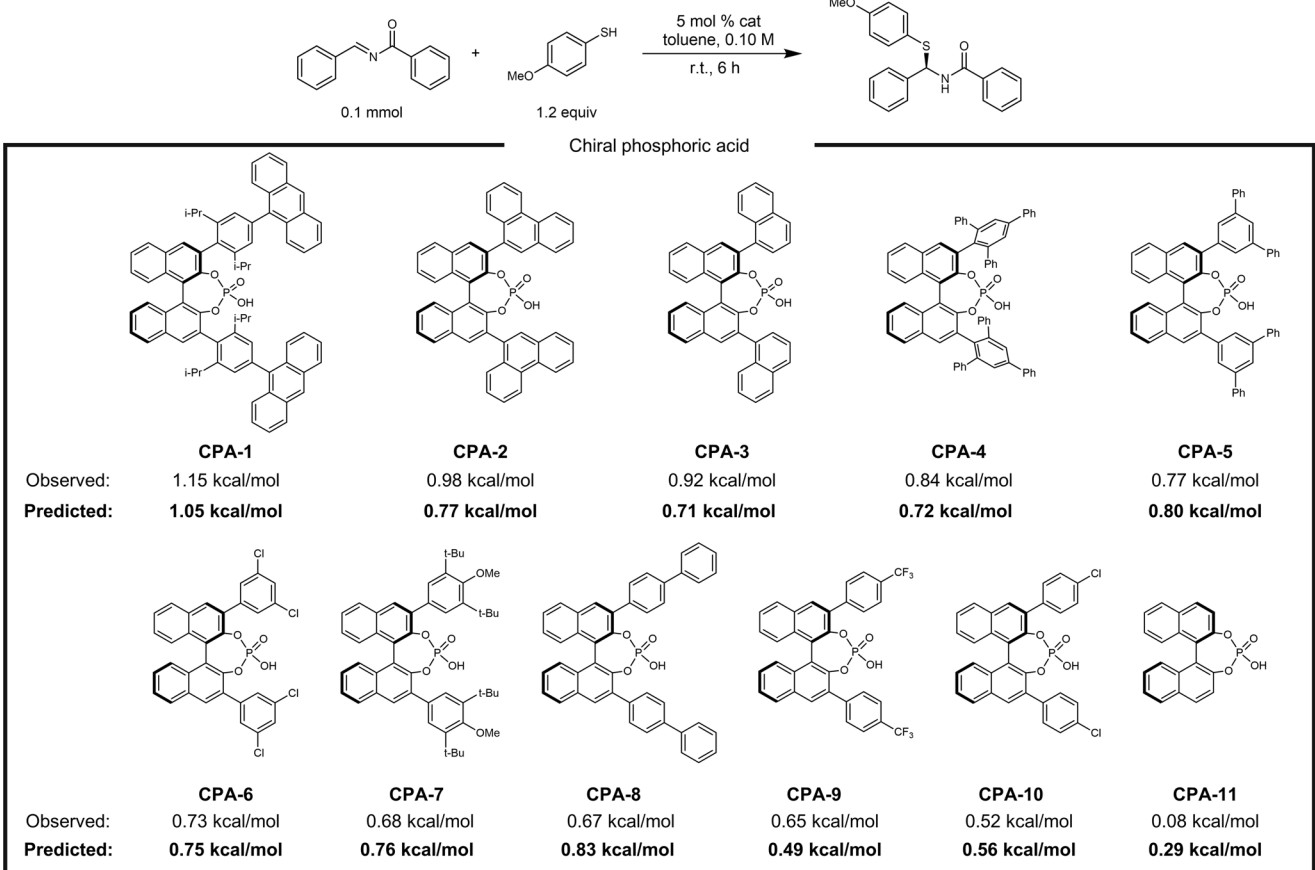

**Fig. 6 | Experimental tests of the extrapolation ability of the SEMG-MIGNN model (steric- and electronics-embedded molecular graph with molecular interaction graph neural network) using additional chiral phosphoric acid catalysts for the asymmetric thiol addition of *N*-acyl amines.** Training with Denmark's dataset, the SEMG-MIGNN model (steric- and electronics-embedded molecular graph with molecular interaction graph neural network) achieved accurate predictions for all the eleven tested phosphoric acids with RMSE (root mean square error) of 0.127 kcal mol⁻¹.

control. Fig. 8b shows the attention weight of electronic encodings at each atom of the representative aryl halides. Comparing the 3-halidepydines (**AH-1, AH-2,** and **AH-3**), the model correctly identified the critical role of halide. When substitution is involved, the attention weights of the *para*-substituted chlorobenzenes revealed the influence of the substituents (**AH-4, AH-5,** and **AH-6**). Although obtained purely from synthetic statistics, the atomic map of electronic influence is consistent with the understanding of organic chemistry and provides the opportunity to harness mechanistic knowledge in synthetic data.

Applying the same interpretation approach to the enantioselectivity prediction model revealed the role of the steric effect on enantioselectivity determination. Fig. 8c shows the changes in predicted enantioselectivity values by eliminating the steric or electronic encodings using **CPA-12** as an example. The perturbations of the enantioselectivity model are in sharp contrast to those of the yield model. Eliminating the electronic encodings lead to very limited influence on the enantioselectivity prediction, 3611 predictions received perturbations within 0.2 kcal mol⁻¹ (Fig. 8c). However, eliminating the steric encodings led to significant changes in the predicted enantioselectivity values (Fig. 8c). This agrees well with the chiral induction model of the asymmetric thiol addition reaction[50] Similarly, the attention weights of the steric encodings can offer insights on the positions related to steric control of the enantioselectivity control (Fig. 8d). In addition to the labeled substituents (i.e., *t*Bu substituent of **CPA-14** and the *ortho*-chloro substituent of **CPA-15**), it is interesting that the model identified the H₈-BINAP backbone (**CPA-12** and **CPA-15**) also contributed to the steric effects.

## Discussion

In summary, we developed a chemical knowledge-based ML model called SEMG-MIGNN for the prediction of synthetic performance. Following the chemical concept, two key designs were implemented: first, the local chemical environment of steric and electronic effect was digitalized and embedded in the graph representation. This significantly enriches the model's characterization of the atomic environment and improves the model's extrapolation ability toward new molecular structures. In addition, an interaction module was developed to enhance the information exchange between reaction components while maintaining the intermolecular boundaries, offering a way to capture the synergistic effect involving multiple reaction components.

The effectiveness of the designed ML model was validated in a number of synthetic prediction tasks with convincing performances. Excellent predictions were found in the yield prediction of Pd-catalyzed C−N cross-coupling reactions and the enantioselectivity prediction of CPA-catalyzed thiol addition to *N*-acyl imines. Further experimental tests of additional CPA catalysts corroborated the model's predictive ability to face unseen molecular structures. Particularly, we found that the SEMG-MIGNN model showed exceptional extrapolative ability in scaffold-based splitting tasks, which is highly desirable for synthetic predictions considering the need to extend the realm of available SPR data.

In addition to the excellent predictive ability, the physically meaningful encodings of steric and electronic effects provide the atomic level of chemical interpretation. Analysis of the trained model revealed the critical role of the electronic effect on the yield

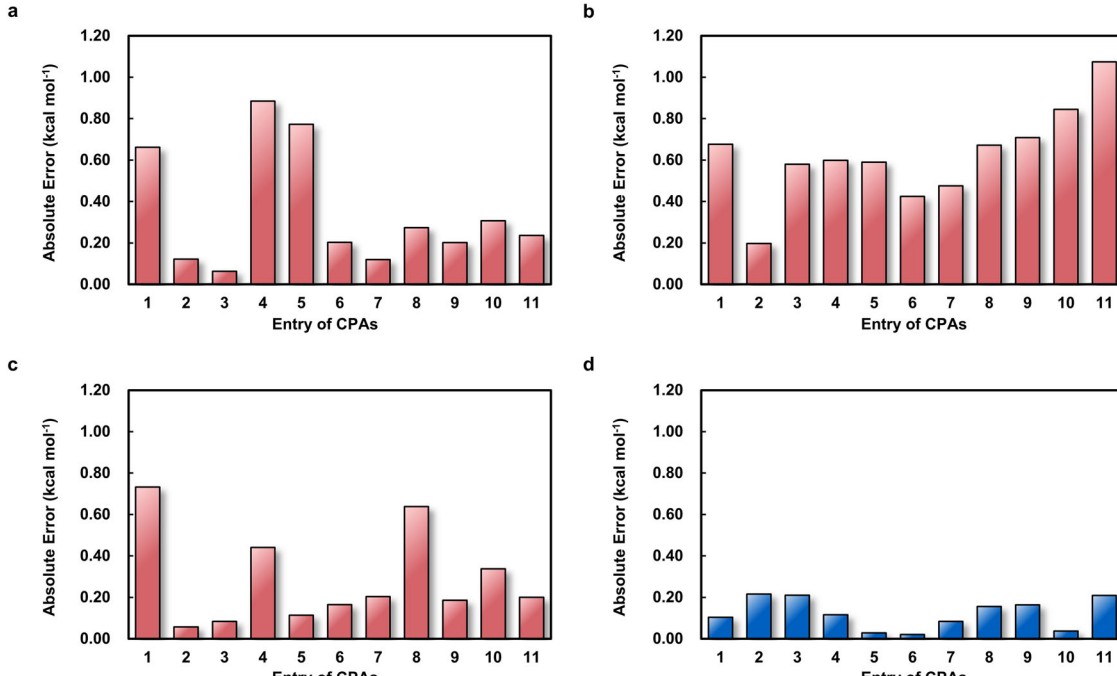

**Fig. 7 | Errors of predictions for the experimentally tested CPAs by SEMG-MIGNN model (steric- and electronics-embedded molecular graph with molecular interaction graph neural network) and other strategies.** SEMG results are shown in blue. Other strategies' results are shown in red. **a** Errors of predictions of the ACSFs-GB model (atom-centered symmetry functions with gradient boosting). **b** Errors of predictions of the DRFP-XGB model (differential reaction fingerprint with XGBoost). **c** Errors of predictions of the MFF-RF model (multiple fingerprint feature with random forest). **d** Errors of predictions of the SEMG-MIGNN model (steric- and electronics-embedded molecular graph with molecular interaction graph neural network). These comparisons demonstrate the advantageous predictions by the SEMG-MIGNN model (steric- and electronics-embedded molecular graph with molecular interaction graph neural network), whose predictions are all within a reasonable error, while the other models encounter pitfall cases where the error of prediction can be larger than 0.5 kcal mol⁻¹. Source data are provided as a Data_for_Fig_7.csv.

prediction of the C−N cross-coupling, while the enantioselectivity prediction relies heavily on the steric effect. Moreover, the model is able to identify the hot spot of the molecular structure for synthetic performance determination, offering useful insight to future designs. The effectiveness of this model shows that integrating representation learning with digitized chemical knowledge can support the development of generalizable models in chemical space, providing the opportunity for the data-driven design of synthetic transformation.

## Methods
### Computation
Details of geometry optimization, standardization of molecular orientation and electron density calculation were elaborated below.

**Step 1: Geometry optimization.** We used RDKit's[51] built-in ETKDG[52] method to generate the initial 3D structure. Subsequent geometry optimizations were performed using the semi-empirical extended tight-binding program package xTB (version 6.3.0), at the GFN2-xTB level of theory. GFN2-xTB default convergence criteria were used. All the GFN2-xTB input and output files are available in our GitHub repository (https://github.com/Shuwen-Li/SEMG-MIGNN).

**Step 2: Standardization of molecular orientation.** Based on the GFN2-xTB-optimized geometry, we standardized the molecular orientation to ensure the consistency of the encodings generated from different initial orientations. For each molecule, we selected three key atoms to determine the orientation of the molecule: the center of gravity, the atom closest to the center of gravity (atom1), and the atom furthest from the center of gravity (atom2). In step 1, the center of gravity is placed at the origin of the *xyz*

coordinate system. In step 2, atom1 is rotated to the positive half of the *z*-axis, which determines the direction of the molecule along the *z*-axis. In step 3, atom2 is rotated to the *yz* plane and placed at the positive half of the *y*-axis, which determines the direction of the molecule along the *y*-axis.

**Step 3: Electron density calculation.** Based on the standardized optimized structures, we used the PySCF package[53] to calculate the electron density distribution of each molecule at the B3LYP/def2-SVP level, which was used for the training of the SEMG-MIGNN model. We also computed the electron densities of other theoretical levels using the PySCF package for benchmark purposes. All the PySCF input and output files are available in our GitHub repository (https://github.com/Shuwen-Li/SEMG-MIGNN).

**SEMG.** The key difference between SEMG and classic molecular graphs is the embedding of steric and electronic information in the atoms (nodes). The detailed workflow of SEMG generation is elaborated below. All the related scripts are available in our GitHub project (https://github.com/Shuwen-Li/SEMG-MIGNN).

**Step 1: Generation of the molecular graph.** Based on the SMILES of the molecule, we obtained the sdf file by RDKit. Using the sdf files, the information on atoms and bonds can be extracted by RDKit. Subsequently, the molecular graph was generated by dgl[54] with the atoms as nodes and the bonds as edges.

**Step 2: Embed steric information.** Based on standardized 3D structure, herein we took the chlorine atom of aryl halides as an example to illustrate the generation of steric information. With the chlorine atom at the center, a sphere of radius 10 Å was built. The radius of this sphere can be customized depending on the sizes of

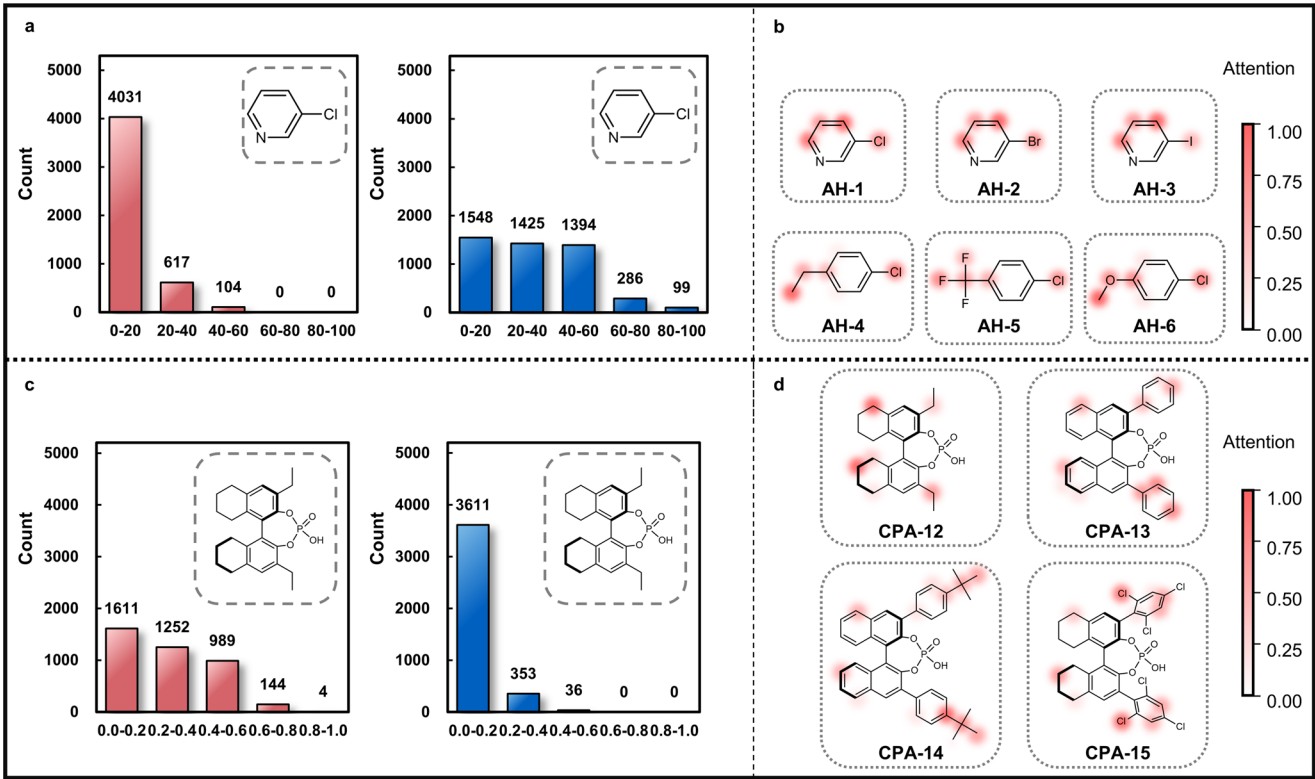

**Fig. 8 | Chemical interpretation of the yield and enantioselectivity determination by the SEMG-MIGNN (steric- and electronics-embedded molecular graph with molecular interaction graph neural network) model. a** Change of predicted yield by eliminating steric (red) or electronic (blue) encodings. **b** Attention weight of electronic encodings at each atom of the selected aryl halides in the yield prediction task. **c** Change of predicted enantioselectivity by eliminating steric (red) or electronic (blue) encodings. **d** Attention weight of steric encodings at each atom of the selected chiral phosphoric acids in the enantioselectivity prediction task. Source data are provided as a Data_for_Fig_8.csv.

the molecules in the dataset. In our study, there were a number of fairly large molecules with more than 180 atoms. 10 Å was assigned as the radius of the sphere to ensure the full description of the steric environment. The distance between the van der Waals surface and the spherical surface was calculated and mapped to the sphere. The steric mapping of the distance on the sphere was encoded as a two-dimensional matrix using equirectangular projection. Based on the selected number $N$ (default is 10) of grids, the polar angle $\theta$ ($0-\pi$) dimension was evenly segmented to $N$ parts ($\theta_i = \pi/N$), and the azimuth angle $\varphi$ ($0-2\pi$) dimension was evenly segmented to $2N$ parts ($\varphi_j = \pi/N$). This provided $N \times 2N$ distance matrix; in our study, a $10 \times 20$ matrix was generated for each atom. This $10 \times 20$ matrix was embedded into the node of the molecular graph to describe the steric information of the atom. Repeating the process for each atom in the molecule, the steric information was embedded in the nodes of the molecular graph.

**Step 3: Embed electronic information.** Starting with the standardized 3D structure, the density matrix was generated by PySCF[53] at the B3LYP/def2-SVP level. Taking the chlorine atom as an example, a $7 \times 7 \times 7$ cubic grid with the chlorine atom at the center was generated. The side length of the cube was the van der Waals diameter of chlorine. According to the Cartesian coordinates of the chlorine atoms and the $7 \times 7 \times 7$ cubic grid, the Cartesian coordinates of the $7 \times 7 \times 7$ cubic grids' center position were obtained. By evaluating the electron density of the $7 \times 7 \times 7$ cubic grids' center position, a $7 \times 7 \times 7$ tensor was generated and further used as the electronic information for embedding into the node of the molecular graph. Repeating the process for each atom in the molecule, the electronic information was embedded in the nodes of the molecular graph.

**Step 4: Generate SEMG.** Eventually, the SEMG was generated as an embedded molecular graph with a $10 \times 20$ steric matrix and $7 \times 7 \times 7$ electronic tensor in each node.

**MIGNN model.** The detailed workflow of the MIGNN model is elaborated as follows. All the related scripts for MIGNN were available in our GitHub project (https://github.com/Shuwen-Li/SEMG-MIGNN).

**Step 1: Input the molecular graphs.** The molecular graphs of all involved compounds were first input into the MIGNN model. The steric information and electronic information were divided into two channels, which are processed separately.

**Step 2: Atom attention.** The steric information and electronic information were processed by several (The number of attention layers was defined by hyper-parameter 'atom_attention'.) attention layers.

**Step 3: Linear layer.** Subsequent processing was several (the number of linear layers was defined by hyper-parameter 'linear_depth'.) linear layers. After the linear layers, they dealt with Batch Normalization and the tan$h$ activation function. In this way, we obtained a steric tensor and an electronic tensor for a given molecule. For a reaction involving $m$ molecules, we would obtain $2m$ different tensors from the above steps, which were $m$ steric tensors and $m$ electronic tensors. Concatenating the $m$ steric tensors, the steric reaction tensor was obtained. Concatenating the $m$ electronic tensors, the electronic reaction tensor was obtained.

**Step 4: Convolution, maxpooling, and flatten layer.** The steric reaction tensor and electronic reaction tensor were processed by

convolution. The steric reaction tensor was put into the 2D convolution layer, and the electronic reaction tensor was put into the 3D convolution layer. Next, the steric reaction tensor passed through maxpooling layer, convolution layer, and flatten layer sequentially, which produced a steric reaction vector. Similarly, the electronic reaction tensor passed through maxpooling layer and flatten layer sequentially, which produced an electronic reaction vector.

**Step 5: Interaction layer.** The steric reaction vector was copied into two parts. For one copy of the steric reaction vector, it was processed by several (the number of attention layers was defined by hyperparameter 'inter_attention') attentions and one linear layer to give a one-dimensional steric representation of the overall transformation. Subsequently, the one-dimensional steric representation and its transpose were multiplied to form the so-called interaction matrix of sterics. The interaction matrix of sterics was then processed by three convolution layers and one flatten layer to produce the interaction vector of sterics. The electronic reaction vector was processed by the same procedures, which led to the interaction vector of electronics.

**Step 6: Concatenating steric information, electronic information, and interaction information.** The replicated steric reaction vector (generated from step 4), the replicated electronic reaction vector (generated from step 4), the interaction vector of sterics (generated from step 5), and the interaction vector of electronics (generated from step 5) were concatenated to the total transformation vector. This total transformation vector includes four parts as elaborated in steps 4 and 5.

**Step 7: Prediction of reaction performance.** From the total transformation vector, we added one attention layer to learn the weight between the total transformation vector and reaction performance. Eventually, the processing of three linear layers led to the predicted reaction performance.

**Interpretation of SEMG-MIGNN.** In order to interpret the trained SEMG-MIGNN model, we evaluate the perturbation of the predicted results by eliminating the steric or electronic encodings for each atom. For a to-be-predicted transformation, we eliminate the steric or electronic encodings of a selected atom in the studying compound of this transformation. This alters the original encodings of the compound to changed encodings with one atom's steric or electronic encodings assigned as zero. Using this changed encoding, the SEMG-MIGNN model would give a different predicted value for the to-be-predicted transformation, and the change of the predicted values (original encoding vs. changed encoding) is the so-called perturbation. Repeating the process for each atom in the molecule, the changes in the predicted values of the molecule were obtained.

**General information about the experiment.** HPLC analysis was performed on Waters-Breeze (2487 Dual λ Absorbance Detector and 1525 Binary HPLC Pump). The Chiralpak OD-H column was purchased from Daicel Chemical Industries, Ltd. Starting materials were purchased from commercial suppliers (Energy Chemical and Adamas-Beta) and used as supplied unless otherwise stated. CPAs were purchased from DAICEL CHIRAL TECHNOLOGIES (CHINA) CO., LTD. Toluene was purified and dried according to standard methods prior to use unless stated otherwise. N-acyl imine was synthesized according to the literature procedure and was purified by sublimation[55].

**General procedure for the reaction of 4-methoxybenzenethiol with *N*-acyl imine.** To a 10 mL vial was added N-acyl imine (20.9 mg, 0.1 mmol) and (R) -CPA* catalyst (5 mol%). Dry toluene (1.0 mL) was added to the mixture, followed by 4-methoxybenzenethiol (0.12 mmol)

via a syringe. The reaction was stirred at room temperature for 30 min. The crude product was purified directly by flash column chromatography (hexane: ethyl acetate = 3:1) to give the corresponding chiral N,S-acetal product. The ee values of the products were determined by chiral HPLC analysis after the products were purified (Daicel Chiralpak OD-H, n-hexane/i-PrOH = 90:10, flow rate = 1.0 mL/min, λ = 231 nm).

**Reporting summary**
Further information on research design is available in the Nature Portfolio Reporting Summary linked to this article.

## Data availability
The authors declare that all the relevant data supporting the findings of this study are available at our repository (https://github.com/Shuwen-Li/SEMG-MIGNN) and Figshare[56]. Source data are provided in this paper.

## Code availability
All codes needed to run this model are available at https://github.com/Shuwen-Li/SEMG-MIGNN[57].

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

## Acknowledgements

National Key R&D Program of China (2022YFA1504301, X.H.), National Natural Science Foundation of China (22122109 and 22271253, X.H.; 22103070, S.-Q.Z.), Zhejiang Provincial Natural Science Foundation of China under Grant No. LDQ23B020002 (X.H.), the Starry Night Science Fund of Zhejiang University Shanghai Institute for Advanced Study (SN-ZJU-SIAS-006, X.H.), Beijing National Laboratory for Molecular Sciences (BNLMS202102, X.H.), CAS Youth Interdisciplinary Team (JCTD-2021-11, X.H.), Fundamental Research Funds for the Central Universities (226-2022-00140, 226-2022-00224 and 226-2023-00115, X.H.), the Center of Chemistry for Frontier Technologies and Key Laboratory of Precise Synthesis of Functional Molecules of Zhejiang Province (PSFM 2021-01, X.H.), the State Key Laboratory of Clean Energy Utilization (ZJUCEU2020007, X.H.), the State Key Laboratory of Physical Chemistry of Solid Surfaces (202210, X. H.), the Leading Innovation Team grant from Department of Science and Technology of Zhejiang Province (2022R01005, X. H.) are gratefully acknowledged. Calculations were performed on the high-performance computing system at the Department of Chemistry, Zhejiang University. We thank Prof. Pu-Sheng Wang for the helpful discussions on experimental verification.

## Author contributions

X.H. and S.-Q.Z. conceived the project and designed the machine learning framework. S.-W.L. performed the machine learning training. L.-C.X. performed training of baseline models. C.Z. performed the experimental verifications. All authors are involved in the discussions and paper writing.

## Competing interests

The authors declare no competing interests.
