## [Peer Review File · Nature Communications]

Reaction performance prediction with an extrapolative and interpretable graph model based on chemical knowledgeREVIEWER COMMENTS

Reviewer #1 (Remarks to the Author):

This work addresses an important topic in a rapidly developing field, and I am generally impressed with the ideas behind the model and the systematic way it has been tested and presented. However, it appears to be mostly an incremental improvement on previous studies and those aspects that at first appear to be a powerful addition, also raise questions about physical accuracy. Furthermore, no direct scientific problem is actually solved in this work, putting even more emphasis on the methodology alone. Overall, I would recommend publication in a journal more specialized in computational methods.

- "For the embedding of the local electronic environment, the initial guess of HF electronic structure calculation by PySCF package³⁷⁻³⁹ is used" - even in the methods section later, it is not at all clear what the authors actually did in this step. I assume HF means Hartree-Fock? then this is a pretty bad reference for the electron density...why not use something more accurate? these are not big systems (even the largest is only 100+ atoms, not a problem for DFT).

- The geometries of the molecules are given by a classical force field MMFF, which is a universal force field and hence at the low end of accuracy even for classical force fields. Again, why not use a more accurate method? with MMFF as an initial guess.

- More generally, I dislike the PySCF package being mentioned as if just the code name itself tells the reader what is being done...it is a large package, with many options and it should be possible to reproduce the calculation as the authors have performed it. I could not find files/data for this aspect in the GitHub repository either. The methods should explain the key aspects and examples should be freely available in the repository.

- If I look at Fig. 4/5, it is clear that the SEMG approach is better, but it is not much better...where is the improvement that makes this method disruptive rather than incremental, justifying publication in Nat. Comm.? I could see an argument in the interpretability of the model, but then the sensitivity of the result to the steric or electronic encodings makes me even more concerned about the low-level of accuracy in the structures. This needs to be benchmarked with higher order methods, although I am missing a compelling reason why the whole thing could not be done with higher order methods, previous work certainly is e.g. see ref 2.

- In Fig. 6, is this actually good enough? the authors state that the worst performer gives an error of 0.3 and the rest are below 0.2, but what is the criteria for being useful in actual reactions? many of the reaction energies are 1 +/- 0.2, so it would not be useful to distinguish between them.

Reviewer #2 (Remarks to the Author):

The study by Li et al. describes the development of a graph neural network model with steric and electronic information (sterics- and electronics-embedded molecular graph, SEMG) for reaction prediction. The SEMG is used in a molecular interaction graph neural network (MIGNN) that is tested on two well-known datasets from Doyle and Denmark, respectively, and show adequate performance in line with previous studies using alternative models (see e.g. 10.1039.D1DD00006C for fingerprints). The authors perform synthesis experiments to augment the Denmark dataset, and show good performance on the new datapoints in comparison with baselines. The models are also interpreted, showing that the steric information is most informative on the stereoselectivity prediction task, while the electronic information is most informative on the yield prediction task, in line with chemical knowledge.

Overall, I like the approach of the paper to add steric and electronic information in a rather unbiased way to the graph neural network model. I believe that it is one way forward to realize better generalization across chemical space and the model and the results are very important and interesting to the community. The paper is well written and I think it will be very understandable to a general chemistry audience. However, there are also issues that call the results into question that would need to be addressed adequately before I would consider it ready for publication. I suggest publication after major revisions.

Major points:

- The pioneering work by Green, Jensen and Coley (10.1039/D0SC04823B, 10.1063/5.0079574) on quantum-chemistry augmented neural networks for reactivity prediction is not cited. This prior work partly negates the authors claims to "this model for the first time embeds the digitalized steric and electronic information of atomic environment". Although these papers only included electronic indices, they need to be discussed extensively as prior work in the field.
- Another prior work on the interaction model is 10.1039/D1SC02087K and in particular the "Reaction GAT". This prior work also needs to be discussed as it partly negates the authors claims of novelty claimed by the authors in "this model for the first time ... the molecular interaction module allows the effective learning of the synergistic control by multiple reaction components"
- Was hyperparameter tuning done for the baseline models employed? There are no details in the paper or the supporting information. From what I can tell from the code repository, the hyperparameters are hardcoded. The baseline models would also need hyperparameter optimization to be adequate baselines. It's not exactly difficult to get really good performance on these two datasets, and I expect the baselines to do even better with hyperparameter tuning.
- Overall, I feel that baselines is a weak point of this manuscript. The whole point is to show that the steric and electronic information is beneficial. Then it needs to be compared to the best available models without this information, for example "yield-BERT" or DRFP from the IBM group, to name a few.
- No consideration is given to the uncertainties in the model scores. As the authors have used Monte Carlo cross-validation with 10 random splits, it is possible to calculate approximate standard errors of the mean for the RMSE, MAE and R2 scores that give some indication whether the studied models are actually significantly different from each other.
- Another important piece of information missing is the learning curves (performance vs amount of data), where the models with steric and electronic information can be expected to perform better with less data (compare approach in 10.1039/D0SC04823B)
- From what I can tell from the repository (e.g., data1_generation-SE.ipynb), min-max scaling is applied to the steric and electronic features before train test split. This is a classic case of data leakage that can cause inflated performance. The models need to be re-run without this data leakage.
- The rotation and translation dependence is fixed with a standardization of the molecular geometry (step 2). It's my impression that the orientation is not uniquely defined by this procedure as the directions of the y and z axes (+-) with respect to the points is not defined.
- Even if the standardization would be uniquely defined, the procedure seems sensitive to the conformer generation procedure. Currently, this is affected by a unset random seed (EmbedMolecule with default arguments). Furthermore, end users might generate their conformers in slightly different ways. I would like to see some sensitivity analysis to how much the predictions are affected by new conformers generated with the same EmbedMolecule function(i.e. different conformers than what the model was trained for).
- The section on extrapolation is currently unconvincing. Although performing additional experiments is a strong point of the paper, there is no assessment of how similar the catalysts are to the ones already in the training set (e.g. fingerprint similarity or more qualitative arguments). I would also like to see some more challenging splits, such as scaffold splits, keeping certain classes of reagents/reactants/catalysts out (see, e.g., 10.1080/1062936X.2021.1883107, 10.1039/D0SC04823B).
- The authors make a big point of their model being able to handle interaction "regardless of the original concatenated sequence". However, this is not shown in the study by e.g. permuting the order for new predictions with a model trained on a fixed order. I would suggest either showing this, or toning down the claims in the paper. My guess is that it would require data augmentation through permutation to achieve this type of invariance.

Minor points:

- The discussion of important targets for ML synthesis prediction (P1, left column) should also include reaction rates/activation energies.
- The authors complain of the high computational cost of QM descriptors (P1, left column), but then go on to calculate their own descriptors based on the electron density and an LDA model. This would surely be on the same time scale as semi-empirical descriptor calculation, so I find the discussion inconsistent with what is actually done in this paper.
- Figure 1, caption: Concatenation is only one of the approaches to handle encoding of multiple molecules with e.g. fingerprints. Two other ones are addition and difference.
- P2, left column: It is said that chemical interpretability of fingerprints is low. I do not agree with this as fingerprints are often interpreted based on their mapping to chemical fragments, see, e.g., 10.1039/D0SC00445F for an example. This interpretability is on par with attention based approaches like the one in the current manuscript.
- P2, left column: It is stated that graph-based predictive models have generated a "strong momentum for artificial intelligence design of functional molecules." I think this is hardly shown in real applications, with the main drawback being the ability to generalize in chemical space.
- The number of significant figures (e.g., 20.662) is surely too large
- Figs. 4 & 5. Note in figure caption which repetition of the train-test split the predictions are from.
- Can the authors please elaborate on the rationale for "eliminating the steric or electronic encodings" by setting them to zero? It is not immediately obvious that setting them to zero would eliminate their influence from the model. The behavior of highly parametrized non-linear models potentially far from any training data input range is unpredictable.
- Figure 8b, lower right: There seems to be attention in the empty space.
- P7, left panel: "This model interpretation offers an interesting chemical knowledge that the electronic effect plays a dominant role for the reaction yield of Pd-catalyzed Buchwald-Hartwig reaction, while the steric effect is limited for the explored reactants. This chemical insight indeed followed the mechanistic understandings and highlighted the interpretability of the SEMG design." This is not really a ground-breaking insight and can be gained from basically any quantum chemical descriptor model.
- P7, left panel: "In addition, the models suggested that the identity of the catalyst backbone (BINAP vs. H8-BINAP) is worth the attention. These interpretation provided valuable hints for further engineering of the chiral phosphoric acid catalysts." These interpretations are also not very informative, as it basically says 'the backbone and the substituents of the catalyst (= the whole catalyst) are important'
- ESI, S5: The rationale for the scaling of Denmark original DD_G values is not clear. If the argument is that there is a racemic background reaction, the influence of this background reaction would depend on the rate of the preferred reaction. This rate is unlikely to be constant over all the studied catalysts. Can the authors please specify their mechanistic model supporting this scaling factor.

Reviewer #3 (Remarks to the Author):

In this manuscript, the author described a novel steric- and electronic-embedded molecular graph (SEMG), the steric environment was generated by spherical projection of molecular stereostructure (SPMS), which was developed by the same group previously. The electronic part was generated by Grid method with the initial guess of HF electronic structure. In addition, a molecular interaction module was also developed to enhance the information exchange between reaction components. The SEMG-MIGNN model showed good performance both in yield and enantioselectivity predictions by using Doyle's C-N cross coupling reaction datasets and Denmark's N,S-acetal formation datasets. Experimental verifications showed good extrapolation ability for this model, model interpretation also gave meaningful information for mechanistic understandings. The results of this manuscript are good, and it may become suitable to be published after the following issues are addressed.

- 1) The author used the MMFF-optimized 3D structure, for simple and rigid molecule, the geometry may be good enough with MMFF force field optimization, however, for large and flexible molecules, the geometries may vary dramatically, how this SEMG-MIGNN model deal with such molecules?
- 2) The molecular interaction module is interesting; however, the interaction matrix was not fully illustrated, how does the interaction of the molecular A and B or A and C was performed in the matrix?
- 3) In the model training part, the author only tested the high through-put dataset such as Doyle and Denmark's reaction dataset, how about the performance with literature-based dataset? The authors have reported an enantioselective hydrogenation dataset, I think this should be also tested with this dataset to validate the robustness of this model. Otherwise, publication is not well justified.
- 4) It seems that the github link does not work, it should be reachable before publication.

Response to Referee 1

Comment: This work addresses an important topic in a rapidly developing field, and I am generally impressed with the ideas behind the model and the systematic way it has been tested and presented. However, it appears to be mostly an incremental improvement on previous studies and those aspects that at first appear to be a powerful addition, also raise questions about physical accuracy. Furthermore, no direct scientific problem is actually solved in this work, putting even more emphasis on the methodology alone. Overall, I would recommend publication in a journal more specialized in computational methods.

Response: We thank the referee for the insightful suggestions and appreciate the recognition of our idea. The key scientific problem targeted in our work is how to integrate the classic knowledge of organic chemistry, namely steric and electronic effects, into chemical machine learning. We believe that the presented SEMG-MIGNN approach provides a general framework for the digitalization of these effects, which leverages the chemical knowledge to improve the AI predictions for reaction performance, leading to improved extrapolative ability and model interpretability.

While there have been a series of exciting advances on the machine learning prediction of molecular synthesis, most existing approaches either use descriptors based on molecular strings or topology, which is challenging to directly reflect the controlling factors of reaction performance, or rely on the hand-picked physical organic parameters that require strong empirical knowledge of the underlying reaction mechanism and structure-performance relationship. *Therefore, the integration of organic knowledge in chemical machine learning remains an open question.* Our work addresses this scientific problem by offering a distinctive approach, with two key differences comparing with previous studies: first, we designed a way to implement the atom-level steric and electronic encodings, which improves the representation of local chemical environment and is applicable to any molecules with well-defined structure; second, we developed a graph neural network to learn the interactions through matrix operations in the modelling process, which effectively captures the complex relationship between reaction components. To further highlight the improving ability of the SEMG-MIGNN approach, we found that our model outperforms the state-of-the-art (SOTA) models in both yield and enantioselectivity tasks, which sets the current SOTA records to the best of our knowledge. We believe that the SEMG-MIGNN model will be applied to tackle challenging synthetic targets and stimulate more works to embed chemical knowledge into advanced AI frameworks, paving the way to promote the intelligent design of chemical reactions.

Regarding the physical accuracy of the computed geometry and electron density, we fully understand the referee's concerns and appreciate the insightful suggestions. We agree that the physical accuracy is the prerequisite to achieve the desired implementation of steric and electronic effects, especially for the aim to train the

chemically aware models. In the revision process, we have carefully studied and addressed the issue of physical accuracy. We systematically compared the MMFF, xTB, and DFT-optimized geometries as well as the electronic densities calculated by thirty-five theoretical methods comprised of five functionals and seven basis sets. Considering both physical accuracy and computational efficiency, we ultimately selected the xTB-optimized geometry and B3LYP/def2-SVP-calculated electron density, which showed significant improvement in physical accuracy compared to the original method (MMFF and Hartree-Fock) and is still suitable for large-scale virtual screening (100,000 to 1,000,000 molecules). Thanks to referee 1's insightful suggestions, we retrained the machine learning models and updated the modelling results.

Comment: "For the embedding of the local electronic environment, the initial guess of HF electronic structure calculation by PySCF package³⁷⁻³⁹ is used" - even in the methods section later, it is not at all clear what the authors actually did in this step. I assume HF means Hartree-Fock? then this is a pretty bad reference for the electron density...why not use something more accurate? these are not big systems (even the largest is only 100+ atoms, not a problem for DFT).

Response: We thank the referee for this important suggestion on the physical accuracy of the electron density calculation. HF refers to Hartree-Fock method, which is a typical low-accuracy method for electron density calculation as the referee suggested. We initially chose the Hartree-Fock method for efficiency reasons; our goal is to allow the applications of the SEMG-MIGNN method in large-scale virtual screening, which typically involves processing molecules in the scale of tens of thousands to millions. However, as the referee pointed out, accuracy is a crucial concern, and physical accuracy is the foundation for the desired digitalization of steric and electronic effects. In light of this critical issue, we have thoroughly evaluated the electron densities calculated by various methods during the revision. We ultimately selected the theoretical level of B3LYP/def2-SVP to obtain the electron densities and process the model trainings.

Based on the GFN2-xTB-optimized geometries (*vide infra*), the accuracies of the computed electron densities were evaluated for thirty-five levels of theory including the variations of five functionals (LDA-VWN, B3LYP, M06-2X, ω B97XD, PBE0) and seven basis sets (STO-3G, STO-6G, 3-21G, def2-SVP, 6-31G(d), 6-311+G**, def2-TZVPP). The evaluation process is elaborated in Figure R1a. For a given molecule in the studied dataset, the electron densities of the same geometry were compared between two levels of theory: the reference level (B3LYP/def2-TZVPP) and the comparing level (the other thirty-four levels). The neighboring electron density of each atom was assessed to obtain a 7x7x7 tensor with the vdW diameter size. This creates a $N \times 7 \times 7 \times 7$ tensor for the entire molecule, which was flattened into an one-dimensional vector. Subsequently, the Euclidean distances between the two vectors were calculated to provide the quantified evaluation of the change of electron densities.

The total of 97 molecules involved in the reactivity and enantioselectivity datasets were examined, and the average Euclidean distances of each level of theory are shown in Figure R1b. This analysis identified four main levels of accuracies for the studied computational methods. As the reviewer pointed out, the previously used Hartree-Fock method was not of the desired accuracy, especially comparing with the recommended DFT level. However, it is also worth noting that as the size of basis set increases, the calculation efficiency decreases significantly. Considering the trade-off between accuracy and efficiency, we have selected the level of B3LYP/def2-SVP (F5B4) for the electron density calculations.

Fig. R1 Quantitative evaluation of the computed electron density at various theoretical levels. **a** Evaluation procedure of the Euclidean distance between the vectors of the computed electron densities. **b** Euclidean distances of the vectors generated by thirty-five theoretical levels.

To further verify the physical accuracy of the selected B3LYP/def2-SVP level, we compared the electrostatic potential surfaces, which is an important representation of the spatial distribution of the electron density. Figure R2 shows the electrostatic potential surfaces of 2-chloropyridine calculated by B3LYP/def2-SVP and B3LYP/def2-TZVPP; under the same scale, the changes between the two levels of theory are quite limited. Detailed comparisons of all 97 molecules are provided in the revised Supplementary Information (Figures S5). These comparisons further demonstrated that the selected B3LYP/def2-SVP approach can provide physically accurate electron density.

Fig. R2 Electrostatic potential surfaces of 2-chloropyridine calculated at the B3LYP/def2-SVP level (a) and the B3LYP/ def2-TZVPP level (b).

In addition, we used the electron densities calculated by the B3LYP/def2-SVP and B3LYP/def2-TZVPP levels to train the SEMG-MIGNN models and compared the prediction performances. Figure R3 shows the model performances trained by different electron density inputs. In both yield and enantioselectivity prediction tasks, further increasing the physical accuracy from def2-SVP level (Figure R3a) to def2-TZVPP level (Figure R3b) only led to limited improvement of regression performances (R^2 : 0.969 vs. 0.971 in yield task; 0.915 vs. 0.918 in enantioselectivity task). These additional evaluations supported that the selected B3LYP/def2-SVP level of theory can provide solid accuracy for the electron density and support the desired machine learning modelling. The related discussions have been included in the revised Supplementary Information.

Fig. R3 Test set performances of the SEMG-MIGNN models trained by the electron density inputs calculated at the B3LYP/def2-SVP level (a) and the B3LYP/def2-TZVPP level (b). The yield dataset is randomly split to 70% (training) and 30% (test). The enantioselectivity task is randomly split to 600 (training) and 475 (test) transformations.

Comment: The geometries of the molecules are given by a classical force field MMFF, which is a universal force field and hence at the low end of accuracy even for classical force fields. Again, why not use a more accurate method? with MMFF as an initial guess.

Response: Thank the referee for the important suggestion on the physical accuracy of molecular geometries. Similar to the electronic density calculation, we initially chose MMFF as the method for geometry optimization due to the considerations of computational efficiency in large-scale virtual screening. When handling datasets

ranging from tens of thousands to millions of molecules (e.g. 133,885 molecules in the QM9 dataset), the cost of geometry optimization is a critical factor that determines the feasibility of the theoretical level. However, as the referee pointed out, the physical accuracy of molecular geometries is also essential for training the desired machine learning model that can comprehend the correct structure-performance relationship instead of simple statistical fitting.

In the revision, we compared the geometries optimized by the MMFF, GFN2-xTB, and DFT (B3LYP/def2-SVP) methods, and representative results are shown in Figure R4. Using the B3LYP/def2-SVP structures as reference, the root-mean-square deviation (RMSD) of the MMFF and GFN2-xTB were computed. Based on these comparisons, we acknowledge that the concerns raised by referee 1 and 3 about the accuracy of MMFF geometries are indeed correct. MMFF is not suitable for the geometry optimization of complex molecules, giving incorrect orientations for certain key substituents (such as the highlighted ones in Figure R4) and yielding an unsatisfying level of RMSD. In comparison, GFN2-xTB significantly improved the accuracy of geometry optimization, achieving a level close to that of DFT optimization while still meeting our requirements for high-throughput virtual screening. Therefore, we eventually chose GFN2-xTB level of theory for geometry optimization, retrained the machine learning models and updated the prediction results.

a

b

Fig. R4 Comparisons of the optimized geometries of representative molecules in yield (a) and enantioselectivity (b) datasets at various levels of theory. RMSDs are computed using the B3LYP/def2-SVP geometries as references.

To ensure the reliability of the GFN2-xTB geometries in terms of modelling accuracy, we further compared the regression performances of SEMG-MIGNN models trained by GFN2-xTB (Figure R5a) and B3LYP/def2-SVP (Figure R5b) geometries. In both yield and enantioselectivity prediction tasks, the two models have comparable predictive abilities. These comparisons further supported that the selected GFN2-xTB level of theory can provide the required accuracy for geometry optimization and enable the desired machine learning modelling. These benchmark results are included in the revised Supplementary Information.

Fig. R5 Test set performances of the SEMG-MIGNN models trained by the geometries optimized at the GFN2-xTB level (a) and the B3LYP/def2-SVP level (b). The yield dataset is randomly split to 70% (training) and 30% (test). The enantioselectivity task is randomly split to 600 (training) and 475 (test) transformations.

Comment: More generally, I dislike the PySCF package being mentioned as if just the code name itself tells the reader what is being done...it is a large package, with many options and it should be possible to reproduce the calculation as the authors have performed it. I could not find files/data for this aspect in the GitHub repository either. The methods should explain the key aspects and examples should be freely available in the repository.

Response: We thank the referee for the valuable feedback and apologize for not providing the complete calculation details and the computational files in the previous version. In the revised Methods section, we have included a full explanation of all the computational details. In addition, all the input and output files involved in this article are now provided in our GitHub repository (<https://github.com/Shuwen-Li/SEMG-MIGNN>), from which the readers can access and reproduce.

Comment: If I look at Fig. 4/5, it is clear that the SEMG approach is better, but it is not much better...where is the improvement that makes this method disruptive rather than incremental, justifying publication in Nat. Comm.? I could see an argument in the interpretability of the model, but then the sensitivity of the result to the steric or electronic encodings makes me even more concerned about the low-level of accuracy in the structures. This needs to be benchmarked with higher order methods, although I am missing a compelling reason why the whole thing could not be done with higher order methods, previous work certainly is e.g. see ref 2.

Response: We appreciate the critical questions raised by the referee. We believe that the key innovation of the proposed SEMG-MIGNN model is that it provides a general framework to encode the steric and electronic effects that chemists are familiar with, thereby providing a new strategy to embed chemical knowledge into the advanced graph AI model and realizing accurate and interpretable reaction modelling.

As the referee suggested, there have already been a series of exciting studies on reaction prediction that have laid a sound foundation for this emerging area. However, previous reaction modelling studies often fell into two extremes of reaction encodings: simple strings/topology-based descriptors or customized physical chemical parameters. While the former approach aims for model generality, it requires strong data support to achieve reliable model training and may have difficulty tracing the physical organic origins. The latter can achieve powerful reactivity and selectivity predictions using very simple regression methods (i.e. multivariate linear regression), but the descriptors are highly customized and require sophisticated understanding of the reaction mechanisms and controlling factors.

In contrast to the above modelling strategies, the core of our work lies in the design of SEMG, which can embed local chemical environments into graph model and provide an effective way to digitalize atomic-level steric and electronic effects. Based on this, we developed the MIGNN framework to enhance the model's learning of molecular

interactions, which is also rarely addressed in previous modelling studies. We believe that these key distinctions provide new ideas for the data-driven modelling of synthetic transformation, and the excellent performances of SEMG-MIGNN model in a series of challenging tasks (*vide infra*) further validated the effectiveness of these designs. This work will stimulate more studies to realize the bridging of chemical knowledge and reaction modelling, paving the way towards AI design of synthetic transformation. We have also elaborated and emphasized these points in the revised manuscript.

Regarding the model performance, we agree with the referee that the previous evaluations did not fully exploit the potential of the SEMG-MIGNN model, and it may seem that the SEMG-MIGNN approach does not offer disruptive prediction ability comparing to the common modelling strategies. Referee 2 also raised the concerns on baseline comparisons and required further evaluations of our model comparing with the SOTA models as well as more challenging application scenarios. During the revision, we evaluated the SEMG-MIGNN model in a series of challenging tasks and demonstrated its powerful predictive ability.

In the yield prediction task, we further compared the SEMG-MIGNN model with several SOTA models, including the MFF encoding by Glorius (*Chem* **6**, 2020, 1379), Reymond's Yield BERT model (*Nat. Mach. Intell* 2021, **3**, 144; *Mach. Learn.: Sci. Technol.* 2021, **2**, 015016) and DRFP model (*Digital Discovery*, 2022, **1**, 91). Table R1 shows the model performances in various data splitting scenarios and additive extrapolation tasks. The SEMG-MIGNN model provides the best performance in most tasks, especially Test 4. These SEMG-MIGNN results also set the current SOTA records for the yield prediction of Buchwald-Hartwig reaction to the best of our knowledge. In addition, the SEMG-MIGNN model demonstrated excellent performances in enantioselectivity prediction and noticeable advancements comparing with the MFF and DRFP models, especially in the extrapolation tasks (Table R2). The predictive ability of the SEMG-MIGNN model were also validated in the experimental tests of chiral phosphoric acids (Figure R6, *vide infra*). These comparisons further demonstrated that the design of SEMG-MIGNN is effective, and the strategy of introducing chemical knowledge in reaction modelling is worth attention for the community.

For the concerns on physical accuracy, we fully agree with the referee's comments and appreciate the insightful suggestions on method benchmarking. The referee is indeed correct. Without physical accuracy, the generated molecular geometries and electronic structures cannot support the desired learning of steric and electronic effects, and thus the predictive reaction modelling cannot be achieved. In the above benchmarking discussions, we carefully evaluated the geometries and electron densities generated by various theoretical levels. Thanks to the referee's suggestion, we were able to choose a significantly more accurate level of theory, GFN2-xTB optimization and B3LYP/def2-SVP electronic structure calculation, to generate the model inputs.

Table R1. Prediction performances of yield tasks using various SOTA models.

Data Splitting	R ² of Yield-BERT	R ² of DRFP	R ² of MFF	R ² of SEMG-MIGNN
Random 90/10	0.962 ± 0.040	0.965 ± 0.010	0.943 ± 0.010	0.970 ± 0.010
Random 80/20	0.957 ± 0.010	0.953 ± 0.005	0.931 ± 0.010	0.971 ± 0.005
Random 70/30	0.952 ± 0.005	0.951 ± 0.005	0.921 ± 0.010	0.969 ± 0.005
Random 60/40	0.934 ± 0.010	0.932 ± 0.010	0.918 ± 0.010	0.963 ± 0.010
Random 50/50	0.922 ± 0.010	0.929 ± 0.010	0.904 ± 0.010	0.941 ± 0.010
Random 40/60	0.901 ± 0.010	0.899 ± 0.010	0.881 ± 0.020	0.921 ± 0.010
Random 30/70	0.883 ± 0.010	0.897 ± 0.010	0.863 ± 0.010	0.903 ± 0.010
Random 20/80	0.862 ± 0.010	0.878 ± 0.010	0.839 ± 0.010	0.883 ± 0.010
Random 10/90	0.791 ± 0.020	0.813 ± 0.010	0.773 ± 0.010	0.834 ± 0.010
Test 1 ^a	0.843 ± 0.010	0.809 ± 0.010	0.853 ± 0.010	0.848 ± 0.010
Test 2 ^a	0.841 ± 0.030	0.832 ± 0.003	0.713 ± 0.005	0.867 ± 0.010
Test 3 ^a	0.753 ± 0.040	0.710 ± 0.001	0.641 ± 0.005	0.776 ± 0.020
Test 4 ^a	0.492 ± 0.050	0.491 ± 0.004	0.178 ± 0.010	0.677 ± 0.020
Avg.1-4	0.732	0.711	0.596	0.792

^aTests 1 to 4 are the extrapolative tests of additives, whose data splitting are determined in Doyle's original study (*Science*, 2018, 360, 186).

Table R2. Prediction performances of enantioselectivity tasks using various SOTA models.

Data Splitting	DRFP		MFF		SEMG-MIGNN	
	R ²	RMSE	R ²	RMSE	R ²	RMSE
Random 90/10	0.903 ± 0.010	0.190 ± 0.010	0.910 ± 0.010	0.183 ± 0.010	0.927 ± 0.005	0.180 ± 0.010
Random 80/20	0.890 ± 0.010	0.223 ± 0.010	0.908 ± 0.010	0.194 ± 0.020	0.929 ± 0.005	0.179 ± 0.010
Random 70/30	0.886 ± 0.010	0.201 ± 0.010	0.895 ± 0.010	0.212 ± 0.020	0.92 ± 0.010	0.189 ± 0.010
Random 60/40	0.873 ± 0.020	0.240 ± 0.020	0.890 ± 0.010	0.230 ± 0.020	0.915 ± 0.010	0.195 ± 0.010
Random 50/50	0.869 ± 0.020	0.248 ± 0.030	0.888 ± 0.020	0.227 ± 0.030	0.905 ± 0.010	0.205 ± 0.020
Random 40/60	0.864 ± 0.020	0.256 ± 0.030	0.885 ± 0.020	0.238 ± 0.030	0.865 ± 0.010	0.245 ± 0.020
Random 30/70	0.863 ± 0.020	0.259 ± 0.030	0.868 ± 0.020	0.243 ± 0.030	0.872 ± 0.010	0.240 ± 0.020
Random 20/80	0.833 ± 0.030	0.286 ± 0.040	0.861 ± 0.030	0.258 ± 0.030	0.834 ± 0.010	0.281 ± 0.020
Random 10/90	0.823 ± 0.020	0.291 ± 0.020	0.776 ± 0.030	0.426 ± 0.030	0.721 ± 0.010	0.370 ± 0.020
Imine ^a	0.904 ± 0.005	0.235 ± 0.005	0.911 ± 0.005	0.226 ± 0.005	0.902 ± 0.005	0.238 ± 0.005
Thiol ^a	-1.585 ± 0.020	0.773 ± 0.020	-1.283 ± 0.020	0.726 ± 0.020	0.611 ± 0.010	0.300 ± 0.010
Catalyst ^a	0.127 ± 0.020	0.561 ± 0.020	0.402 ± 0.020	0.464 ± 0.020	0.753 ± 0.010	0.298 ± 0.010
Transformation ^a	0.885 ± 0.005	0.233 ± 0.005	0.835 ± 0.005	0.264 ± 0.005	0.912 ± 0.005	0.205 ± 0.005

^aThese data splitting tasks refer to the extrapolative predictions based on the scaffold splitting of the reaction components. Details are elaborated in Figure R7. RMSEs are in kcal/mol.

Comment: In Fig. 6, is this actually good enough? the authors state that the worst performer gives an error of 0.3 and the rest are below 0.2, but what is the criteria for being useful in actual reactions? many of the reaction energies are 1 +/- 0.2, so it would not be useful to distinguish between them.

Response: Regarding the accuracy requirement for the enantioselectivity prediction in actual reactions, the ee values typically range between 0% to 99%. This corresponds to a $\Delta\Delta G$ value within 3.2 kcal/mol at room temperature. Therefore, we believe that the accuracy of 0.2 kcal/mol is sufficient to provide chemically meaningful predictions for actual enantioselective transformations. This is also consistent with the current SOTA records for the machine learning of Denmark's dataset. For random data splitting, Denmark's original study has a mean absolute error (MAE) of 0.152 kcal/mol (*Science* **363**, 2019, eaau5631), and Glorius' MFF model can achieve a MAE of 0.144 kcal/mol (*Chem* **6**, 2020, 1379).

For the 11 additional CPA tests in Fig. 6, the distribution of $\Delta\Delta G$ values is indeed fairly narrow as the referee pointed out (0.08 kcal/mol to 1.15 kcal/mol). This is because the commercially available CPAs are limited; removing the 43 CPAs that are already present in Denmark's dataset, there are quite few CPAs left which can be purchased and have the structural diversity to support the extrapolative challenges. This is the major reason for the narrow distribution of $\Delta\Delta G$ values produced by the tested CPAs.

However, we want to emphasize that the extrapolative tests of the 11 CPAs are still statistically meaningful. Figure R6 compares the regression performances of the SEMG-MIGNN model with other SOTA models (DRFP and MFF). The SEMG-MIGNN model achieved a R^2 of 0.745 and a RMSE of 0.127 kcal/mol; while the accuracies of the MFF model and the DRFP model are significantly worse, with R^2 values of -0.845 and -5.207 respectively. These comparisons further highlighted the accuracy of the SEMG-MIGNN model in the extrapolation tasks, which is highly desirable in chemical predictions. Based on the SEMG-MIGNN predictions, if we were to choose the best catalyst prior to the experimental screening, the top-1 candidate **CPA-1** is indeed the correct selection with a 0.1 kcal/mol prediction error, which further support that the model can provide useful differentiation of the candidate CPAs.

Fig. R6 Extrapolative enantioselectivity predictions of new chiral phosphoric acids by the DRFP, MFF and SEMG-MIGNN models.

Response to Referee 2

Comment: The study by Li et al. describes the development of a graph neural network model with steric and electronic information (sterics- and electronics-embedded molecular graph, SEMG) for reaction prediction. The SEMG is used in a molecular interaction graph neural network (MIGNN) that is tested on two well-known datasets from Doyle and Denmark, respectively, and show adequate performance in line with previous studies using alternative models (see e.g. 10.1039.D1DD00006C for fingerprints). The authors perform synthesis experiments to augment the Denmark dataset, and show good performance on the new datapoints in comparison with baselines. The models are also interpreted, showing that the steric information is most informative on the stereoselectivity prediction task, while the electronic information is most informative on the yield prediction task, in line with chemical knowledge.

Overall, I like the approach of the paper to add steric and electronic information in a rather unbiased way to the graph neural network model. I believe that it is one way forward to realize better generalization across chemical space and the model and the results are very important and interesting to the community. The paper is well written and I think it will be very understandable to a general chemistry audience. However, there are also issues that call the results into question that would need to be addressed adequately before I would consider it ready for publication. I suggest publication after major revisions.

Response: We appreciate the referee's valuable feedbacks and are grateful for the opportunity to improve our work. We share the same view that the digitalization of chemical knowledge and its unbiased embedding in machine learning modelling is a useful step towards the improvement of generalization ability across the chemical space. Such models will provide opportunity to make chemically meaningful and valuable predictions that can guide and inspire new chemical discoveries. We hope the design of SEMG-MIGNN can provide a helpful reference architecture for the community, together pushing the digital transformation of synthetic chemistry.

We also take the suggestions on the background introduction and model evaluation seriously. During the revision, we carefully revised and expanded the discussion of prior arts. These discussions properly acknowledge the important contributions made by the pioneering researchers in this field and place our work in the appropriate context. In addition, we have conducted thorough examinations of the SEMG-MIGNN model in a wide range of challenging scenarios and compared its performance with the SOTA models. To our delight, the SEMG-MIGNN model outperforms the SOTA models in the majority of tasks for both yield and enantioselectivity prediction. Thanks to the referee's suggestion, these additions have significantly improved our work and further validated the effectiveness of the SEMG-MIGNN design.

Comment: The pioneering work by Green, Jensen and Coley (10.1039/D0SC04823B, 10.1063/5.0079574) on quantum-chemistry augmented neural networks for reactivity prediction is not cited. This prior work partly negates the authors claims to "this model for the first time embeds the digitalized steric and electronic information of atomic environment". Although these papers only included electronic indices, they need to be discussed extensively as prior work in the field.

Response: We appreciate the referee's insightful suggestions. We completely agree that the pioneering studies by Green, Jensen, Coley and co-workers on quantum-chemistry augmented neural networks should be discussed extensively in our paper. These landmark studies (10.1039/D0SC04823B cited as ref. 17 in the original version and 10.1063/5.0079574 cited as ref. 42 in the revised version) provide an innovative strategy for incorporating electronic information into graph neural networks in a site-specific fashion, which significantly improved the prediction of regioselectivity and reactivity.

In the revised manuscript, we have included a detailed discussion of the related studies on the embedding of chemical information in graph models. Particularly, we have highlighted the key advances which inspired the design of the SEMG-MIGNN model, as well as discussed our innovations. These additional discussions provided a more comprehensive introduction to the prior arts and placed our work in the appropriate context.

Comment: Another prior work on the interaction model is 10.1039/D1SC02087K and in particular the "Reaction GAT". This prior work also needs to be discussed as it partly negates the authors claims of novelty claimed by the authors in "this model for the first time ... the molecular interaction module allows the effective learning of the synergistic control by multiple reaction components"

Response: Thank the referee for this important reminder. We appreciate the insightful suggestion and agree with the necessity to discuss the prior work from Liu, Yu, and co-workers (10.1039/D1SC02087K). Their "Reaction GAT" module in DeepReac+ connects the GAT-processed molecular vectors through a hyper-graph, allowing the model to enhance information interaction between reaction components. Although our MIGNN model differs in terms of modelling approach, both works aim to address the same chemical issue. We have carefully revised the relevant discussions and emphasized the excellent solutions provided by Liu, Yu and co-workers.

Comment: Was hyperparameter tuning done for the baseline models employed? There are no details in the paper or the supporting information. From what I can tell from the code repository, the hyperparameters are hardcoded. The baseline models would also need hyperparameter optimization to be adequate baselines. It's not exactly difficult to get really good performance on these two datasets, and I expect the baselines to do even better with hyperparameter tuning.

Response: Following the referee's suggestion, we have further conducted hyperparameter optimization for the baseline models during the revision. For the GCN model, the range of the hyperparameter optimization included: convolution layer = [1, 2, 3], multi graph = [mean, sum, max], output = [mean, sum, max]. For the MIGNN model, the range of the hyperparameter optimization included: linear_depth = [0,1,2,3,4,5,6,7,8,9,10], hidden_size = [8,16,32,64,128,256], atom_attention = [0,1,2], inter_attention = [0,1,2], end_attention = [0,1,2], fc_size = [8,16,32,64,128,256], and final_act = ['sigmoid', 'none'].

As a result of the hyperparameter optimization, the baseline models showed improved performance, but still remained a certain gap from the SEMG-MIGNN. Table R3 summarizes the changes in model performance prior to and after the hyperparameter optimization. For the yield prediction task, SEMG-MIGNN has the best performance with R^2 of 0.970 after the hyperparameter optimization. SEMG-MIGNN also achieved the highest R^2 of 0.915 in the enantioselectivity task. Additionally, we also conducted the hyperparameter optimization for the common molecular descriptors (OH, RDKit, MF, ACSFs) to ensure the comparisons at the same level. The details of the hyperparameter optimization have been clarified in the revised Supplementary Information.

Table R3. Modelling performances in yield and enantioselectivity tasks using baseline models prior to and after the hyperparameter optimization. The yield dataset is randomly split to 70% (training) and 30% (test). The enantioselectivity task is randomly split to 600 (training) and 475 (test) transformations.

Prediction Task	Descriptor	Model	R^2 Prior to Optimization	R^2 After Optimization
Yield	OH	XGBoost	0.911	0.912
Yield	RDKit	XGBoost	0.930	0.934
Yield	MF	XGBoost	0.930	0.938
Yield	ACSFs	XGBoost	0.923	0.929
Yield	Baseline MG	GCN	0.423	0.546
Yield	SEMG	GCN	0.559	0.591
Yield	Baseline MG	MIGNN	0.917	0.920
Yield	SEMG	MIGNN	0.952	0.970
Enantioselectivity	OH	RandomForest	0.885	0.885
Enantioselectivity	RDKit	Gradient Boosting	0.900	0.900
Enantioselectivity	MF	Gradient Boosting	0.900	0.901
Enantioselectivity	ACSFs	Gradient Boosting	0.896	0.900
Enantioselectivity	Baseline MG	GCN	0.361	0.777
Enantioselectivity	SEMG	GCN	0.802	0.815
Enantioselectivity	Baseline MG	MIGNN	0.838	0.876
Enantioselectivity	SEMG	MIGNN	0.903	0.915

Comment: Overall, I feel that baselines is a weak point of this manuscript. The whole point is to show that the steric and electronic information is beneficial. Then it needs to be compared to the best available models without this information, for example "yield-BERT" or DRFP from the IBM group, to name a few.

Response: Thank the referee for this important suggestion. We fully agree that comparing SEMG-MIGNN to SOTA models, especially in the challenging prediction tasks, is crucial for the evaluation of the SEMG-MIGNN design and verifying the idea of incorporating the steric and electronic information into graph model. In the revision, we compared the SEMG-MIGNN model with the best available models in a series of yield and enantioselectivity prediction tasks. These models include the Yield BERT and DRFP as suggested by the referee, as well as the recently developed MFF approach by Glorius (*Chem* **6**, 2020, 1379). To our delight, SEMG-MIGNN outperforms the SOTA models in the majority of tasks for both yield and enantioselectivity predictions.

Table R4 compares the performance of SOTA models in yield prediction. We tested 13 prediction tasks, including different ratios of random data splitting and extrapolative predictions for 4 additives. In the random data splitting, SEMG-MIGNN outperformed the other tested models in all nine tasks. For the extrapolative predictions of the additives, SEMG-MIGNN achieved the best performance in tests 2 to 4, and MFF provided better predictions in test 1 (R^2 of 0.853, MFF; R^2 of 0.848, SEMG-MIGNN). It is worth noting that for the challenging extrapolative predictions, SEMG-MIGNN showed a noticeable advantage, especially in Test 4. These results further demonstrated that the SEMG-MIGNN model has exceptional ability in yield prediction.

We further compared the SOTA models in 13 enantioselectivity prediction tasks. In addition to the 9 random data splitting tasks with different ratios of training data, we also divided the imines, thiols, and catalysts based on the molecular scaffold, thereby producing the extrapolative challenges according to the referee's suggestion. Figure R7 elaborates the details of these scaffold-based data splitting. The division of imines classified imine-5 with bicyclic naphthyl substituent as the test set, while only monocyclic aryl substituents were included in the training set. Thiols were classified to aromatic thiols (training set) and aliphatic thiols (test set). For the phosphoric acid catalysts, they were divided to the training set of BINAP CPAs and the test set of H₈-BINAP CPAs. In addition to these scaffold-based splitting, we also examined the transformation-based splitting; the 9 transformations involving imine-1 and thiol-1 were divided to the test set, while the remaining 16 transformations were used as the training set. The above data splitting posed a series of extrapolative challenges for the machine learning models and examined the prediction performances under application scenarios.

Table R4. Prediction performances of yield tasks using various SOTA models.

Data Splitting	R ² of Yield-BERT	R ² of DRFP	R ² of MFF	R ² of SEMG-MIGNN
Random 90/10	0.962 ± 0.040	0.965 ± 0.005	0.943 ± 0.010	0.970 ± 0.005
Random 80/20	0.957 ± 0.010	0.953 ± 0.005	0.931 ± 0.010	0.971 ± 0.005
Random 70/30	0.952 ± 0.005	0.951 ± 0.005	0.921 ± 0.010	0.969 ± 0.005
Random 60/40	0.934 ± 0.010	0.932 ± 0.010	0.918 ± 0.010	0.963 ± 0.010
Random 50/50	0.922 ± 0.010	0.929 ± 0.010	0.904 ± 0.010	0.941 ± 0.010
Random 40/60	0.901 ± 0.010	0.899 ± 0.010	0.881 ± 0.020	0.921 ± 0.010
Random 30/70	0.883 ± 0.010	0.897 ± 0.010	0.863 ± 0.010	0.903 ± 0.010
Random 20/80	0.862 ± 0.010	0.878 ± 0.010	0.839 ± 0.010	0.883 ± 0.010
Random 10/90	0.791 ± 0.020	0.813 ± 0.010	0.773 ± 0.010	0.834 ± 0.010
Test 1 ^a	0.843 ± 0.010	0.809 ± 0.010	0.853 ± 0.010	0.848 ± 0.010
Test 2 ^a	0.841 ± 0.030	0.832 ± 0.003	0.713 ± 0.005	0.867 ± 0.010
Test 3 ^a	0.753 ± 0.040	0.710 ± 0.001	0.641 ± 0.005	0.776 ± 0.020
Test 4 ^a	0.492 ± 0.050	0.491 ± 0.004	0.178 ± 0.010	0.677 ± 0.020
Avg.1-4	0.732	0.711	0.596	0.792

^aTests 1 to 4 are the extrapolative tests of additives, whose data splitting are determined in Doyle's original study (*Science*, 2018, 360, 186).

Table R5 summarized the performances of DRFP, MFF, and SEMG-MIGNN models in the enantioselectivity prediction tasks. Yield-BERT was not considered because it was not developed for enantioselectivity prediction. Our model presented noticeable improvements in most scenarios. In the random data splitting, only in the case of very limited training data (10% and 20% training data), SEMG-MIGNN is worse than DRFP or MFF. While in the other random data splitting scenarios, SEMG-MIGNN outcompeted the DRFP and MFF models.

Particularly noteworthy is the extrapolative predictions, where the improvement of SEMG-MIGNN is evident. In the extrapolations of thiol and catalyst, DRFP and MFF showed poor or even incorrect predictions, while the predictions of SEMG-MIGNN are still competent without pitfall scenarios. For the transformation-out splitting, SEMG-MIGNN is also the only model with a R² over 0.9. These results demonstrated that the SEMG-MIGNN model also has the desired performance in enantioselectivity prediction, especially in challenging extrapolation tasks.

Fig. R7 Data splitting of Denmark's enantioselectivity dataset based on molecular scaffolds and transformations.

Table R5. Prediction performances of enantioselectivity tasks using various SOTA models.

Data Splitting	DRFP		MFF		SEMG-MIGNN	
	R ²	RMSE	R ²	RMSE	R ²	RMSE
Random 90/10	0.903 ± 0.010	0.190 ± 0.010	0.910 ± 0.010	0.183 ± 0.010	0.927 ± 0.005	0.180 ± 0.010
Random 80/20	0.890 ± 0.010	0.223 ± 0.010	0.908 ± 0.010	0.194 ± 0.020	0.929 ± 0.005	0.179 ± 0.010
Random 70/30	0.886 ± 0.010	0.201 ± 0.010	0.895 ± 0.010	0.212 ± 0.020	0.920 ± 0.010	0.189 ± 0.010
Random 60/40	0.873 ± 0.020	0.240 ± 0.020	0.890 ± 0.010	0.230 ± 0.020	0.915 ± 0.010	0.195 ± 0.010
Random 50/50	0.869 ± 0.020	0.248 ± 0.030	0.888 ± 0.020	0.227 ± 0.030	0.905 ± 0.010	0.205 ± 0.010
Random 40/60	0.864 ± 0.020	0.256 ± 0.030	0.885 ± 0.020	0.238 ± 0.030	0.865 ± 0.010	0.245 ± 0.010
Random 30/70	0.863 ± 0.020	0.259 ± 0.030	0.868 ± 0.020	0.243 ± 0.030	0.872 ± 0.010	0.240 ± 0.010
Random 20/80	0.833 ± 0.030	0.286 ± 0.040	0.861 ± 0.030	0.258 ± 0.030	0.834 ± 0.010	0.281 ± 0.010
Random 10/90	0.823 ± 0.020	0.291 ± 0.020	0.776 ± 0.030	0.426 ± 0.030	0.721 ± 0.010	0.370 ± 0.010
Imine ^a	0.904 ± 0.005	0.235 ± 0.005	0.911 ± 0.005	0.226 ± 0.005	0.902 ± 0.005	0.238 ± 0.005
Thiol ^a	-1.585 ± 0.020	0.773 ± 0.020	-1.283 ± 0.020	0.726 ± 0.020	0.611 ± 0.010	0.300 ± 0.010
Catalyst ^a	0.127 ± 0.020	0.561 ± 0.020	0.402 ± 0.020	0.464 ± 0.020	0.753 ± 0.010	0.298 ± 0.010
Transformation ^a	0.885 ± 0.005	0.233 ± 0.005	0.835 ± 0.005	0.264 ± 0.005	0.912 ± 0.005	0.205 ± 0.005

^aThese data splitting tasks refer to the extrapolative predictions based on the scaffold splitting of the reaction components. Details are elaborated in Figure R7. RMSEs are in kcal/mol.

Comment: No consideration is given to the uncertainties in the model scores. As the authors have used Monte Carlo cross-validation with 10 random splits, it is possible to calculate approximate standard errors of the mean for the RMSE, MAE and R² scores that give some indication whether the studied models are actually significantly different from each other.

Response: We have now included the approximate standard errors of the RMSE, MAE and R² scores for the Monte Carlo cross-validations. These results have been updated in the revised manuscript and Supplementary Information, providing additional insights into the model uncertainties.

Comment: Another important piece of information missing is the learning curves (performance vs amount of data), where the models with steric and electronic information can be expected to perform better with less data (compare approach in 10.1039/D0SC04823B)

Response: We have further tested the performance of the SEMG-MIGNN model with different amounts of training data. Figure R8 shows the model's learning curves. In both yield and enantioselectivity tasks, the SEMG-MIGNN model can achieve an acceptable performance with 20% of the training data, and its predictive ability approached convergence with 70% or more the training data. Related details have been included in the revised Supplementary Information.

Fig. R8 Learning curves of the SEMG-MIGNN model in yield (a) and enantioselectivity (b) tasks.

Comment: From what I can tell from the repository (e.g., data1_generation-SE.ipynb), min-max scaling is applied to the steric and electronic features before train test split. This is a classic case of data leakage that can cause inflated performance. The models need to be re-run without this data leakage.

Response: We thank the referee for this important suggestion. We did apply min-max scaling to the steric and electronic features before the data splitting, which indeed caused data leakage as pointed out by the referee. In the revision, we have re-trained the models without the normalization and compared the results. Figure R9 showed the modelling results with the incorrect normalization and without normalization. Indeed, the prediction results have a marginal improvement with the incorrect normalization of the entire dataset. We have removed the normalization and updated all the modelling results in the revision.

Fig. R9 Modelling performances with or without the unified normalization. a Results in yield prediction task (random 70/30). b Results in enantioselectivity prediction task (random 600/475).

Comment: The rotation and translation dependence is fixed with a standardization of the molecular geometry (step 2). It's my impression that the orientation is not uniquely defined by this procedure as the directions of the y and z axes (+/-) with respect to the points is not defined.

Response: Thank the referee for pointing out the missing details in our description of the standardization procedure. Our code determined the direction of the y and z axes during the standardization process. We have clarified the details in the Methods section to avoid misunderstandings.

Figure R10, using cyclohexylthiol as an example, illustrates the workflow of the standardization process. For each molecule, we selected three key atoms to determine the orientation of the molecule: the center of gravity, the atom closest to the center of gravity (atom1), and the atom furthest from the center of gravity (atom2). In step 1, the center of gravity is placed at the origin of the xyz coordinate system. In step 2, atom1 is rotated to the positive half of the z-axis, which determines the direction of the molecule along the z-axis. In step 3, atom2 is rotated to the yz plane and placed at the positive half of the y-axis, which determines the direction of the molecule along the y-axis. These three steps standardize the orientation of the molecule, ensuring the consistency of the encodings generated from different initial coordinates.

Fig. R10 Standardization procedure of the molecular orientation using cyclohexylthiol as an example.

Comment: Even if the standardization would be uniquely defined, the procedure seems sensitive to the conformer generation procedure. Currently, this is affected by a unset random seed (EmbedMolecule with default arguments). Furthermore, end users might generate their conformers in slightly different ways. I would like to see some sensitivity analysis to how much the predictions are affected by new conformers generated with the same EmbedMolecule function(i.e. different conformers than what the model was trained for).

Response: In the revision, we further tested the impact of the initial structure on the modelling performance. Ten different random seeds were applied for the generation of the molecular structures using the EmbedMolecule module. Subsequently, we performed the geometry optimizations and electronic structure calculations through the same process. The changes in optimized structures and prediction performances are summarized in Table R6 (yield task) and Table R7 (enantioselectivity task), which showed marginal influence from the selection of random seed. These additional results demonstrate that the model is not sensitive to the initial random seed. Related discussions are included in the revised Supplementary Information. The corresponding random seeds are also provided on my GitHub repository (<https://github.com/Shuwen-Li/SEMG-MIGNN>) for readers to reproduce.

Table R6. Structural RMSDs and modelling performances in yield prediction task (70% training and 30% test) using different random seeds for the generation of initial structure.

Seed	Averaged RMSD (Å) ^a	RMSE (%)	R ²
1	--	4.88	0.968
2	1.12	4.59	0.972
3	0.99	4.79	0.969
4	1.12	4.41	0.975
5	1.17	5.19	0.964
6	1.18	4.79	0.969
7	1.20	4.50	0.974
8	1.23	4.85	0.968
9	1.16	5.11	0.965
10	1.19	4.71	0.970

^aThe structural RMSDs were determined using the structures of seed 1 as the reference. The computed RMSDs of 44 molecules are averaged.

Table R7. Structural RMSDs and modelling performances in enantioselectivity prediction task (600 training and 475 test) using different random seeds for the generation of initial structure.

Seed	Averaged RMSD (Å) ^a	RMSE (kcal/mol)	R ²
1	--	0.199	0.912
2	2.79	0.206	0.907
3	2.71	0.196	0.915
4	2.93	0.199	0.913
5	2.74	0.190	0.918
6	2.95	0.203	0.909
7	2.92	0.186	0.922
8	2.93	0.199	0.913
9	2.93	0.205	0.906
10	2.78	0.195	0.916

^aThe structural RMSDs were determined using the structures of seed 1 as the reference. The computed RMSDs of 53 molecules are averaged.

Comment: The section on extrapolation is currently unconvincing. Although performing additional experiments is a strong point of the paper, there is no assessment of how similar the catalysts are to the ones already in the training set (e.g. fingerprint similarity or more qualitative arguments). I would also like to see some more challenging splits, such as scaffold splits, keeping certain classes of reagents/reactants/catalysts out (see, e.g., 10.1080/1062936X.2021.1883107, 10.1039/D0SC04823B).

Response: Thank the referee for this important suggestion. We fully agree that the quantification of the structural differences between the Denmark dataset and the newly tested CPAs, as well as additional challenging splits, are crucial for demonstrating the model's extrapolative ability.

Following the referee's suggestion, we used the correlation coefficient of Morgan molecular fingerprints to evaluate the structural differences between the 43 CPAs in Denmark's dataset and the 11 CPAs we tested experimentally. The distribution of the correlation coefficients is shown in Figure R11a. These results indicated that the CPAs in our experimental evaluations have noticeable differences in terms of the topological structure. The median value of the correlation coefficient is 0.56, whose structures are shown in Figure R11b. Related discussions are included in the revised Supplementary Information.

Fig. R11 Quantifying the structural similarities between the experimentally tested CPAs and the CPAs in Denmark's dataset. **a.** Distribution of the correlation coefficients in Morgan molecular fingerprints. **b.** The chemical structure of the pair of CPAs that have the median value of correlation coefficient.

In addition, we have re-divided the datasets of yield and enantioselectivity from the perspectives of scaffold splitting and transformation-out, and compared the SEMG-MIGNN model with other SOTA models in a series of prediction tasks. Figure R12 shows the details of scaffold splitting in the yield dataset. For aryl halides, the substituted arenes were selected in the training set, and the pyridines were included in the test set. For Buchwald ligands, we chose the two ligands with the additional methoxy substituent as the training set and the rest two ligands as the test set. For base, the guanidine-type organic bases are used for the training set, and phosphazene are included in the test set. For the oxazole additives, we selected the mono-substituted ones as the training set and the di-substituted ones as the test set. The above scaffold-based splittings have clear organic chemistry meanings and pose extrapolative challenges from the synthetic perspective. We also want to note that the transformation-out splitting cannot be performed on Doyle's yield dataset, due to the fact that there is only one molecule (*p*-toluidine) as the N-coupling partner.

Figure R13 summarizes the results of the extrapolation tasks for yield prediction using the SEMG-MIGNN model and other recommended SOTA models (Yield-BERT, DRFP, and MFF). SEMG-MIGNN model demonstrated noticeable advantage. The arene-to-pyridine extrapolation task of aryl halides is the most difficult among the four extrapolation challenges; SEMG-MIGNN achieved a regression performance with R^2 of 0.576, which is significantly higher than the R^2 of the other three models (0.230, Yield-BERT; 0.222, DRFP; 0.449, MFF). In the extrapolation tasks for additive, ligand, and base, the tested SOTA models also did not achieve satisfying regression performances, with R^2 ranging from 0.3 to 0.5, making it difficult to provide synthetically useful predictions. However, our SEMG-MIGNN model achieved R^2 of 0.851 in the additive task, 0.816 in the ligand task, and 0.658 in the base task. These results further demonstrated that the SEMG-MIGNN design has exceptional ability for the extrapolative prediction tasks and can effectively transfer the structure-performance relationship between molecular scaffolds.

Fig. R12 Scaffold splitting of Doyle's yield dataset.

Fig. R13 Modelling results in the scaffold-based extrapolation tasks of yield prediction using SEMG-MIGNN and other SOTA models. **a** Summary of the regression performances. **b** Regression performances of Yield-BERT model. **c** Regression performances of DRFP model. **d** Regression performances of MFF model. **e** Regression performances of SEMG-MIGNN model.

We also performed the scaffold-based splitting on Denmark's enantioselectivity dataset and examined the model performances for the extrapolation tasks. Figure R14 elaborates the details of the extrapolation splittings. The division of imines classified imine-5 with bicyclic naphthyl substituent as the test set, while only monocyclic aryl substituents were included in the training set. Thiols were classified to aliphatic thiols (test set) and aromatic thiols (training set). For the phosphoric acid catalysts, they were divided to the training set of BINAP CPAs and the test set of H₈-BINAP CPAs. In addition to these scaffold-based splitting, we also examined the transformation-based splitting; the 9 transformations involving imine-1 and thiol-1 were divided to the test set, while the remaining 16 transformations were used as the training set.

Fig. R14 Data splitting of Denmark's enantioselectivity dataset based on molecular scaffolds and transformations.

Fig. R15 Modelling results in the extrapolation tasks of enantioselectivity prediction using SEMG-MIGNN and other SOTA models. **a** Summary of the regression performances. **b** Regression performances of DRFP model. **c** Regression performances of MFF model. **d** Regression performances of SEMG-MIGNN model. RMSEs are in kcal/mol.

We compared the SEMG-MIGNN model with other SOTA models (DRFP, and MFF) in the extrapolation tasks for the enantioselectivity prediction, and the results are summarized in Figure R15. SEMG-MIGNN model outperforms the other tested models in most cases except the imine extrapolation. For the extrapolation of imines, all the evaluated models can provide satisfying regression performance (R^2 higher than 0.90), and the DRFP and MFF model has the best result ($R^2 = 0.911$). For the thiol

extrapolation task, it is noteworthy that the DRFP and MFF models would fail, giving incorrect predictions with negative R^2 ; while the SEMG-MIGNN model can still provide reasonable predictions with a R^2 of 0.611 and RMSE of 0.300 kcal/mol. Similar situation exists for the catalyst extrapolation. The regression with DRFP model only achieved a $R^2 = 0.116$, and that with MFF model has a $R^2 = 0.402$. In contrast, the predictions with SEMG-MIGNN model showed significant improvement in the catalyst extrapolation, with $R^2 = 0.753$ and RMSE of 0.298 kcal/mol. For the transformation-out splitting, the SEMG-MIGNN model still has the highest regression performance with a $R^2 = 0.912$. We also want to note that the predictions in the experimentally tested CPAs also confirmed that the SEMG-MIGNN model outcompetes the DRFP and MFF models with noticeable improvement (R^2 of 0.745, SEMG-MIGNN; R^2 of -5.207, DRFP; R^2 of -0.845, MFF; Figure R6). These extrapolation challenges provide strong support for the predictive ability of the SEMG-MIGNN model, which have been included in the revised manuscript.

Comment: The authors make a big point of their model being able to handle interaction "regardless of the original concatenated sequence". However, this is not shown in the study by e.g. permuting the order for new predictions with a model trained on a fixed order. I would suggest either showing this, or toning down the claims in the paper. My guess is that it would require data augmentation through permutation to achieve this type of invariance.

Response: We are grateful for bringing this important issue to our attention. The referee is correct that our previous statements on the concatenated sequence were misleading. The original discussions were only meant to emphasize that all possible combinations of reaction components have been represented in the matrix operation, which avoid the need for strict requirements of data formatting in reaction modelling (i.e. reactant-catalyst-product). However, a unified data format is still necessary throughout the modelling process. As a result, our model currently does not have the ability to provide the same predictions with permuted order. The referee's suggestion regarding this matter is correct, and we have revised the related discussions in the manuscript.

Comment: The discussion of important targets for ML synthesis prediction (P1, left column) should also include reaction rates/activation energies.

Response: Following the referee's suggestion, we have included the discussions on reaction rates and activation energies in the revised manuscript. This information certainly enriches our discussion on ML synthesis prediction.

Comment: The authors complain of the high computational cost of QM descriptors (P1, left column), but then go on to calculate their own descriptors based on the electron density and an LDA model. This would surely be on the same time scale as semi-empirical descriptor calculation, so I find the discussion inconsistent with what is actually done in this paper.

Response: We agree with the referee that our model still requires certain computational cost, for which the previous discussions may mislead the readers. We have corrected the related discussions in the revised manuscript.

Comment: Figure 1, caption: Concatenation is only one of the approaches to handle encoding of multiple molecules with e.g. fingerprints. Two other ones are addition and difference.

Response: We have revised Figure 1's caption to accurately reflect the available approaches for handling multiple molecules.

Comment: P2, left column: It is said that chemical interpretability of fingerprints is low. I do not agree with this as fingerprints are often interpreted based on their mapping to chemical fragments, see, e.g., 10.1039/D0SC00445F for an example. This interpretability is on par with attention based approaches like the one in the current manuscript.

Response: We thank the referee for this important notice. We agree that the fingerprints can be interpreted to provide mechanistic insights, as described in the suggested reference. We have made the revisions to accurately reflect the chemical interpretability of fingerprints.

Comment: P2, left column: It is stated that graph-based predictive models have generated a "strong momentum for artificial intelligence design of functional molecules." I think this is hardly shown in real applications, with the main drawback being the ability to generalize in chemical space.

Response: We appreciate the referee's insight and agree that there are still challenges for the graph-based models to achieve the desired ability to generalize in chemical space. We have revised the related discussions.

Comment: The number of significant figures (e.g., 20.662) is surely too large

Response: We have checked and corrected the number of significant figures throughout the manuscript and Supplementary Information.

Comment: Figs. 4 & 5. Note in figure caption which repetition of the train-test split the predictions are from.

Response: We have revised the figure captions to clarify the details of the data splitting.

Comment: Can the authors please elaborate on the rationale for "eliminating the steric or electronic encodings" by setting them to zero? It is not immediately obvious that setting them to zero would eliminate their influence from the model. The behavior of highly parametrized non-linear models potentially far from any training data input range is unpredictable.

Response: We understand the referee's concern about the model analysis of the steric and electronic contributions. For highly parameterized non-linear models, there lacks a universally accepted way to decompose the fragmental contributions. Setting the direct mathematical outcome of the concerned encodings to zero is a working approach to quantify the perturbation of the prediction results, which offers useful information on the trained structure-performance model. We think this is a reasonable way to evaluate how much the steric and electronic encodings can perturb the prediction results, which is mechanistically relevant.

However, as the referee pointed out, this does not mean that the perturbation quantifies the overall contributions of the steric or electronic encodings, since their sum is not the total prediction value. We also clarified the related discussions to avoid misunderstandings.

Comment: Figure 8b, lower right: There seems to be attention in the empty space.

Response: Thank the referee for this notice. We have corrected Figure 8b for this mistake.

Comment: P7, left panel: "This model interpretation offers an interesting chemical knowledge that the electronic effect plays a dominant role for the reaction yield of Pd-catalyzed Buchwald-Hartwig reaction, while the steric effect is limited for the explored reactants. This chemical insight indeed followed the mechanistic understandings and highlighted the interpretability of the SEMG design." This is not really a ground-breaking insight and can be gained from basically any quantum chemical descriptor model.

Response: We agree with the referee that other physical parameters with clear steric and electronic definitions would probably reach the same conclusions. We have toned down the related discussions to avoid over-emphasizing the model interpretation.

Comment: P7, left panel: "In addition, the models suggested that the identity of the catalyst backbone (BINAP vs. H8-BINAP) is worth the attention. These interpretation provided valuable hints for further engineering of the chiral phosphoric acid catalysts." These interpretations are also not very informative, as it basically says 'the backbone and the substituents of the catalyst (== the whole catalyst) are important'

Response: We agree with the referee that the interpretation may not be that straightforward as reflected by the high-dimensional nature of the structure-enantioselectivity relationship. Perhaps the most useful way of catalyst engineering is to perform high-throughput virtual screening with the trained model. We have revised related discussions to avoid over-emphasizing the attention-based interpretation.

Comment: ESI, S5: The rationale for the scaling of Denmark original DD_G values is not clear. If the argument is that there is a racemic background reaction, the influence of this background reaction would depend on the rate of the preferred reaction. This rate is unlikely to be constant over all the studied catalysts. Can the authors please specify their mechanistic model supporting this scaling factor.

Response: We thank the referee for this important question. The influence of the racemic background reaction, probably catalyzed by the trace amount of inseparable Lewis acid, is indeed related to the rate of the CPA-catalyzed transformation. During the experimental explorations, we noticed that the imine addition is very fast, and we tried our best to eliminate the influence of background reaction. We have tried various means of purification and more stringent reaction setups, such as new glassware for each transformation, but we still cannot completely avoid the reduction of enantioselectivity (95% ee in our repetition for the reference CPA) comparing to Denmark's study (99% ee, *Science* **363**, 2019, eaau5631). Nonetheless, our scaling factor is a reasonable compromise. The mechanistic reasoning is elaborated as follows:

Without the background reaction, $\Delta\Delta G^\ddagger = -RT\ln\frac{1+ee}{1-ee} = -RT[\ln(1+ee) - \ln(1-ee)]$

With the background reaction, $\Delta\Delta G_{w/bg}^\ddagger = -RT[\ln(1+ee_{w/bg}) - \ln(1-ee_{w/bg})]$

Applying second-order Taylor expansion:

$$\begin{aligned}\frac{\Delta\Delta G^\ddagger}{\Delta\Delta G_{w/bg}^\ddagger} &= \frac{ee - \frac{1}{2}ee^2 - \left(-ee - \frac{1}{2}ee^2\right)}{ee_{w/bg} - \frac{1}{2}ee_{w/bg}^2 - \left(-ee_{w/bg} - \frac{1}{2}ee_{w/bg}^2\right)} = \frac{ee}{ee_{w/bg}} \\ &= \frac{k_{pref_tot} + k_{bg_tot}}{k_{pref_tot}} = 1 + \frac{k_{bg_tot}}{k_{pref_tot}}\end{aligned}$$

Because the substrates were synthesized by the same reaction and subsequently purified, the content of the Lewis acid should be comparable, and the corresponding background reaction rate is approximately constant (k_{bg_tot}). In addition, since all the chiral catalysts are CPA, the total rates of the catalytic reactions (k_{pref_tot}) should also be approximately comparable (k_{pref_tot} is the total reaction rate under CPA catalysis, which does not require the enantioselectivity to be the same). Therefore, even there is a discrepancy in the actual enantioselectivity, due to the limited fluctuation of the catalytic reaction rate, we believe the $\frac{\Delta\Delta G^\ddagger}{\Delta\Delta G_{w/bg}^\ddagger}$ can be considered as a constant because of the above approximation.

Response to Referee 3

Comment: In this manuscript, the author described a novel steric- and electronic-embedded molecular graph (SEMG), the steric environment was generated by spherical projection of molecular stereostructure (SPMS), which was developed by the same group previously. The electronic part was generated by Grid method with the initial guess of HF electronic structure. In addition, a molecular interaction module was also developed to enhance the information exchange between reaction components. The SEMG-MIGNN model showed good performance both in yield and enantioselectivity predictions by using Doyle's C-N cross coupling reaction datasets and Denmark's N,S-acetal formation datasets. Experimental verifications showed good extrapolation ability for this model, model interpretation also gave meaningful information for mechanistic understandings. The results of this manuscript are good, and it may become suitable to be published after the following issues are addressed.

Response: Thank the referee for the valuable comments and suggestions. We appreciate the opportunity to revise the manuscript.

Comment: The author used the MMFF-optimized 3D structure, for simple and rigid molecule, the geometry may be good enough with MMFF force field optimization, however, for large and flexible molecules, the geometries may vary dramatically, how this SEMG-MIGNN model deal with such molecules?

Response: Thank the referee for the important suggestion on the physical accuracy of molecular geometries. We initially chose MMFF as the method for geometry optimization due to the considerations of computational efficiency in large-scale virtual screening. However, as the referee pointed out, the physical accuracy of molecular geometries is also essential for training the desired machine learning model that can comprehend the correct structure-performance relationship instead of simple statistical fitting.

In the revision, we compared the geometries optimized by the MMFF, GFN2-xTB, and DFT (B3LYP/def2-SVP) methods, and representative results are shown in Figure R16. Using the B3LYP/def2-SVP structures as reference, the root-mean-square deviation (RMSD) of the MMFF and GFN2-xTB were computed. Based on these comparisons, we acknowledge that the concerns raised by referee about the accuracy of MMFF geometries are indeed correct. MMFF is not suitable for the geometry optimization of complex molecules, giving incorrect orientations for certain key substituents (such as the highlighted ones in Figure R16) and yielding an unsatisfying level of RMSD. In comparison, GFN2-xTB significantly improved the accuracy of geometry optimization, achieving a level close to that of DFT optimization while still meeting our requirements for high-throughput virtual screening. Therefore, we eventually chose GFN2-xTB level of theory for geometry optimization, retrained the machine learning models and updated the prediction results.

Fig. R16 Comparisons of the optimized geometries of representative molecules in yield (a) and enantioselectivity (b) datasets at various levels of theory. RMSDs are computed using the B3LYP/def2-SVP geometries as references.

To ensure the reliability of the GFN2-xTB geometries in terms of modelling accuracy, we further compared the regression performances of SEMG-MIGNN models trained by GFN2-xTB (Figure R17a) and B3LYP/def2-SVP (Figure R17b) geometries. In both yield and enantioselectivity prediction tasks, the two models have comparable predictive abilities. These comparisons further supported that the selected GFN2-xTB level of theory can provide the required accuracy for geometry optimization and enable the desired machine learning modelling. These benchmark results are included in the revised Supplementary Information.

To ensure the reliability of the ML results, we also tested the impact of the initial structure on the modelling performance. Ten different random seeds were applied for the generation of the molecular structures using the EmbedMolecule module. Subsequently, we performed the geometry optimizations and electronic structure calculations through the same process. The changes in optimized structures and prediction performances are summarized in Table R8 (yield task) and Table R9 (enantioselectivity task), which showed marginal influence from the selection of random seed. These additional results demonstrate that the revised model is not sensitive to initial conformation. Related discussions are included in the revised Supplementary Information.

Fig. R17 Test set performances of the SEMG-MIGNN models trained by the geometries optimized at the GFN2-xTB level (a) and the B3LYP/def2-SVP level (b). The yield dataset is randomly split to 70% (training) and 30% (test). The enantioselectivity task is randomly split to 600 (training) and 475 (test) transformations.

Table R8. Structural RMSDs and modelling performances in yield prediction task (70% training and 30% test) using different random seeds for the generation of initial structure.

Seed	Averaged RMSD(Å) ^a	RMSE (%)	R ²
1	--	4.88	0.968
2	1.12	4.59	0.972
3	0.99	4.79	0.969
4	1.12	4.41	0.975
5	1.17	5.19	0.964
6	1.18	4.79	0.969
7	1.20	4.50	0.974
8	1.23	4.85	0.968
9	1.16	5.11	0.965
10	1.19	4.71	0.970

^aThe structural RMSDs were determined using the structures of seed 1 as the reference.

Table R9. Structural RMSDs and modelling performances in enantioselectivity prediction task (600 training and 475 test) using different random seeds for the generation of initial structure.

Seed	Averaged RMSD(Å) ^a	RMSE (kcal/mol)	R ²
1	--	0.199	0.912
2	2.79	0.206	0.907
3	2.71	0.196	0.915
4	2.93	0.199	0.913
5	2.74	0.190	0.918
6	2.95	0.203	0.909
7	2.92	0.186	0.922
8	2.93	0.199	0.913
9	2.93	0.205	0.906
10	2.78	0.195	0.916

^aThe structural RMSDs were determined using the structures of seed 1 as the reference.

Comment: The molecular interaction module is interesting; however, the interaction matrix was not fully illustrated, how does the interaction of the molecular A and B or A and C was performed in the matrix?

Response: Our design aims to achieve the information interaction through matrix multiplication. Figure R18 illustrates the details of the interaction matrix. The encodings of each reaction component are concatenated to create the one-dimensional matrix expression X (or an undirected vector) of the total transformation. The matrix X and its transpose matrix X^T are combined to form the interaction matrix, which includes the interactions between each reaction component. Taking the interaction between A and B as an example, the highlighted two matrices are the multiplication of the one-dimensional matrix corresponding to A and the transpose of the one-dimensional matrix corresponding to B ($A \times B^T$), and the multiplication of the one-dimensional matrix corresponding to B and the transpose of the one-dimensional matrix corresponding to A ($B \times A^T$). The related details have also been added in the revised Methods section.

Fig. R18 Details of the matrix operations in the design of interaction module.

Comment: In the model training part, the author only tested the high through-put dataset such as Doyle and Denmark's reaction dataset, how about the performance with literature-based dataset? The authors have reported an enantioselective hydrogenation dataset, I think this should be also tested with this dataset to validate the robustness of this model. Otherwise, publication is not well justified.

Response: We appreciate the important question raised by the referee. We fully agree that additional model evaluations are crucial to justify the design of SEMG-MIGNN. Following the referee's suggestion, we further compared the performance of SEMG-MIGNN with other SOTA models (MFF, *Chem* **6**, 2020, 1379; DRFP, *Digital Discovery*, 2022, **1**, 91) on the asymmetric hydrogenation of olefins. The representative Rh/BINOL-phosphite-catalyzed hydrogenation reaction of tri-substituted olefins was studied, and the 10-fold cross validation performances are compared in Figure R19. To our delight, SEMG-MIGNN also exhibited satisfying prediction performance in this transformation, with a R^2 of 0.777 and a RMSE of 0.381 kcal/mol, which outperforms DRFP and MFF approaches. Additionally, in light of referee 2's suggestions, we have also validated the predictive ability of SEMG-MIGNN model on additional challenging extrapolation tasks (Figures R12 to R15). These results collectively provide strong support for the effectiveness of the SEMG-MIGNN design.

Fig. R19 Enantioselectivity prediction of Rh/BINOL-phosphite-catalyzed hydrogenation reaction of tri-substituted olefins using SEMG-MIGNN and other SOTA models. The models were trained using 10-fold cross validation.

Comment: It seems that the github link does not work, it should be reachable before publication.

Response: Thank the referee for bringing this issue to our attention. We have updated the content on our GitHub repository and verified that the link (<https://github.com/Shuwen-Li/SEMG-MIGNN>) is now valid.

REVIEWERS' COMMENTS

Reviewer #1 (Remarks to the Author):

I think the authors have made a comprehensive effort to answer the comments of all the referees and would be happy for the manuscript to proceed to publication.

Reviewer #2 (Remarks to the Author):

I would like to congratulate the authors on the extremely extensive revisions to the manuscript, which have significantly improved it and addressed all my previous concerns. It is rare to see such well-considered revisions and such a well-written response letter and I applaud that. In particular the very good results for extrapolation tasks is a notable advance compared to previous methods. I strongly recommend that the paper should be published in Nature Communications and it will be of great interest to the community.

Minor point: Please clarify what the "correlation coefficients" are that were used to calculate the structural differences between the new CPAs and those in the Denmark dataset. Normally this is done with the Tanimoto similarity.

Reviewer #3 (Remarks to the Author):

The revised manuscript by Hong and co-workers addressed the issues we pointed before, the new models were based on more accurate DFT calculated structures and the model performance was further improved. I appreciate the effort the authors made to introduce chemical knowledge-based sterics and electronics information into the MG model, I think the manuscript is well organized and suitable to be published in Nat. Commun.

Response to Referee 1

Comment: I think the authors have made a comprehensive effort to answer the comments of all the referees and would be happy for the manuscript to proceed to publication.

Response: We appreciate the referee's recognition of our work. The professional suggestion significantly improved the quality of our research. We hope that the design of SEMG-MIGNN can offer a useful method to address challenges in synthetic chemistry.

Response to Referee 2

Comment: I would like to congratulate the authors on the extremely extensive revisions to the manuscript, which have significantly improved it and addressed all my previous concerns. It is rare to see such well-considered revisions and such a well-written response letter and I applaud that. In particular the very good results for extrapolation tasks is a notable advance compared to previous methods. I strongly recommend that the paper should be published in Nature Communications and it will be of great interest to the community.

Response: We are grateful for the referee's appreciation and recognition of our research. Thanks for the valuable comments and suggestions, which have been helpful in guiding us to improve our work.

Comment: Please clarify what the "correlation coefficients" are that were used to calculate the structural differences between the new CPAs and those in the Denmark dataset. Normally this is done with the Tanimoto similarity.

Response: We thank the referee for this important notice. The structural differences between the tested CPAs and those in Denmark's dataset are evaluated by the correlation coefficient of Tanimoto similarity using Morgan molecular fingerprints. To avoid misunderstanding, we add the details of it in manuscript and supplementary information.

Response to Referee 3

Comment: The revised manuscript by Hong and co-workers addressed the issues we pointed before, the new models were based on more accurate DFT calculated structures and the model performance was further improved. I appreciate the effort the authors made to introduce chemical knowledge-based sterics and electronics information into the MG model, I think the manuscript is well organized and suitable to be published in Nat. Commun.

Response: We are grateful for the referee's support and recognition of our work. Thanks

for the insightful suggestions, which have helped us to improve the quality of our work.